# Gene-specific nonsense-mediated mRNA decay targeting for cystic fibrosis therapy

Young Jin Kim [1,2,3], Tomoki Nomakuchi[1,3,5], Foteini Papaleonidopoulou [1,6], Lucia Yang [1,2,3], Qian Zhang[1,4] & Adrian R. Krainer [1✉]

Low *CFTR* mRNA expression due to nonsense-mediated mRNA decay (NMD) is a major hurdle in developing a therapy for cystic fibrosis (CF) caused by the W1282X mutation in the *CFTR* gene. CFTR-W1282X truncated protein retains partial function, so increasing its levels by inhibiting NMD of its mRNA will likely be beneficial. Because NMD regulates the normal expression of many genes, gene-specific stabilization of *CFTR*-W1282X mRNA expression is more desirable than general NMD inhibition. Synthetic antisense oligonucleotides (ASOs) designed to prevent binding of exon junction complexes (EJC) downstream of premature termination codons (PTCs) attenuate NMD in a gene-specific manner. We describe cocktails of three ASOs that specifically increase the expression of *CFTR*-W1282X mRNA and CFTR protein upon delivery into human bronchial epithelial cells. This treatment increases the CFTR-mediated chloride current. These results set the stage for clinical development of an allele-specific therapy for CF caused by the W1282X mutation.

[1] Cold Spring Harbor Laboratory, Cold Spring Harbor, NY 11724, USA. [2] Graduate Program in Genetics, Stony Brook University, Stony Brook, NY 11794, USA. [3] Medical Scientist Training Program, Stony Brook University School of Medicine, Stony Brook, NY 11794, USA. [4] Graduate Program in Molecular and Cell Biology, Stony Brook University, Stony Brook, NY 11794, USA. [5] Present address: Division of Human Genetics, Children's Hospital of Philadelphia, Philadelphia, PA 19104, USA. [6] Present address: Francis Crick Institute, London 1140062, UK. ✉email: krainer@cshl.edu

CFTR-W1282X, the 6th most common CF-causing mutation, causes a severe form of CF and is present in 1.2% of CF patients worldwide[1], 2.2% of U.S. CF patients[2], and up to 40% of Israeli CF patients[3]. The CFTR-W1282X truncated protein retains partial function[4–6], but is expressed at a very low level, due to nonsense-mediated mRNA decay (NMD). In general, NMD prevents the accumulation of potentially harmful truncated proteins translated from premature termination codon (PTC)-containing mRNAs. However, when NMD reduces the expression of a mutant CFTR protein that has partial activity, it exacerbates the phenotype, so that patients homozygous for the CFTR-W1282X mutation or compound heterozygous for CFTR-W1282X and another CF-causing mutation have poor clinical outcomes. It has been estimated that 10–30% of normal CFTR function provides a significant therapeutic benefit for CF patients[7,8]. Thus, increasing the expression of mutant CFTR protein with residual activity is expected to be beneficial.

NMD is a major hurdle for developing a targeted therapy for CF caused by the CFTR-W1282X mutation. The approval of CFTR correctors that enhance post-translational CFTR processing, and potentiators that improve CFTR channel opening, brought benefit to the majority of CF patients[9]. However, these therapeutic options are not effective against CF caused by CFTR-W1282X, due to the low expression of CFTR-W1282X mRNA. One approach to treat CF caused by this mutation involves read-through compounds (RTCs) that increase the level of full-length protein by reducing the fidelity of the ribosome at the PTC[10]. Gentamicin is a type of RTC that can increase full-length CFTR protein in vitro, but its clinical efficacy for various CF nonsense mutations is limited by NMD[11,12]. Likewise, ataluren is a non-aminoglycoside RTC with a very good safety profile, but it did not improve forced expiratory volume (FEV) in CF patients with various nonsense mutations, including W1282X, in clinical trials[13].

The efficacy of RTCs can be increased in vivo by knocking down key components of the NMD pathway[14]. However, a clinically viable NMD-suppression approach does not exist yet. CFTR potentiators and correctors such as ivacaftor (VX-770) and lumacaftor (VX-809) enhance CFTR-W1282X activity in vitro, and may potentially benefit patients with the W1282X mutation, but they are not effective if the truncated protein expression is too low[4–6,15]. Thus, there is a pressing need for strategies to overcome NMD of the CFTR-W1282X mRNA.

Several NMD-suppression strategies have been developed for potential application to diseases caused by NMD-sensitive nonsense mutations. These include inhibition of NMD by small-molecule inhibitors[16,17] or knockdown of key NMD factors[18,19]. However, global inhibition of NMD may be detrimental, because the NMD machinery targets a subset of normal and physiologically functional mRNA isoforms, thereby post-transcriptionally regulating gene expression[20]. Therefore, global inhibition of NMD could disrupt mRNA homeostasis in a broad range of tissues[14].

Uniformly modified antisense oligonucleotides (ASOs) can stably hybridize to complementary RNAs without triggering their RNase-H-mediated degradation[21,22]. Such ASOs are effective tools for disrupting the interaction between an RNA and its binding proteins, and can be used to alter mRNA processing or translation in vitro and in vivo[21,22]. A strategy for overcoming NMD of CFTR-W1282X mRNA involves such ASOs: splice-switching antisense oligonucleotides (ASOs) composed of phosphorodiamidate morpholino oligomers or 2′-O-(2-methoxyethyl) (MOE)-modified ribose and phosphorothioate backbone (PS), and designed to induce skipping of the PTC-containing exon 23 promote expression of a partially active CFTR variant lacking the amino acids encoded by the skipped exon[23–25].

An alternative gene-specific NMD inhibition strategy we report here involves targeting the binding of a complex of RNA-binding proteins called the exon junction complex (EJC), a key component of the NMD pathway. In contrast to many RNA-binding proteins, EJCs bind mRNA in a position-dependent, sequence-independent manner[26,27]. More than 80% of EJCs are positioned 20~24 nucleotides (nt) upstream of an exon-exon junction[26,27]. The '55-nt rule' predicts that mRNAs with a PTC > 55 nt upstream of the last exon-exon junction are degraded by NMD, reflecting the footprint of the stalled ribosome[20]. A downstream EJC interacts with the ribosome stalled at the PTC, and recruits NMD factors to form a degradation complex that promotes decapping, deadenylation, and endocleavage of the target mRNA, which is subsequently degraded[20]. Thus, the EJC is a major enhancer of NMD.

Disrupting the downstream EJC association with PTC-containing mRNA can be used to inhibit NMD[28]. We previously developed ASOs that target presumptive downstream EJC sites of PTC-containing mRNAs. These ASOs efficiently attenuate NMD of their target genes, restoring mRNA and proteins levels[28]. In the present study, we demonstrate that cocktails of three ASOs targeting presumptive downstream EJC binding sites specifically increase the expression of endogenous CFTR-W1282X mRNA in human bronchial epithelial (HBE) cells. Furthermore, the ASO cocktails increase partially active CFTR protein and CFTR-mediated chloride current in HBE cells. These results set the stage for the clinical development of an allele-specific therapy for CF caused by the W1282X mutation.

## Results

**ASO screening with minigene NMD reporters**. CFTR-W1282X mRNA has four downstream exon-exon junctions, on exons 23–24, 24–25, 25–26, and 26–27. However, the predicted EJC binding site on exon 23 is approximately 5 nt downstream of the PTC—within the ribosome footprint—and thus only three exons (24, 25, and 26) are predicted to harbor EJCs that can induce NMD. To investigate the impact of each EJC on NMD of CFTR-W1282X mRNA, we generated U2OS cells stably expressing doxycycline-inducible CFTR-minigene NMD reporters, each with only one presumptive downstream EJC site (Fig. 1a, Supplementary Fig. 1a). These NMD reporters are three-exon minigenes downstream of GFP, and comprise CFTR cDNA sequence for exons 22–27 and intervening sequences (IVSs) that are shortened natural CFTR introns.

The reporters showed some intron retention, and pW1282X-IVS23 generated an additional isoform by use of an alternative 3′ splice site (3′ss; based on size and motif predictions) (Supplementary Fig. 1b–f). The '55-nt rule' predicts that an exon-exon junction <55 nt downstream of the PTC does not induce NMD[20]. As mentioned above, only the EJCs on exon 24, 25, and 26 are predicted to induce NMD. Inhibiting NMD with cycloheximide increased the mRNA levels of the reporters harboring the PTC and intron 24, 25, or 26 (pW1282X-IVS24, pW1282X-IVS25, or pW1282X-IVS26) (Supplementary Fig. 1d–f); conversely, the reporters harboring the PTC and intron 23 (pW1282X-IVS23) or harboring intron 24 but not the PTC (pWT-IVS24) were not sensitive to cycloheximide (Supplementary Fig. 1b, c).

Uniformly MOE-modified ASOs can stably hybridize to complementary mRNAs without inducing RNAse-H-mediated degradation, and modulate their posttranscriptional processing, including NMD[21]. Using our previously described ASO screening strategy for gene-specific antisense inhibition of NMD—dubbed "GAIN"[28]— we designed sets of 19 overlapping 15-mer ASOs to target each of the presumptive EJC binding sites on the NMD reporters pW1282X-IVS24, pW1282X-IVS25, and pW1282X-IVS26, respectively (Fig. 1b).

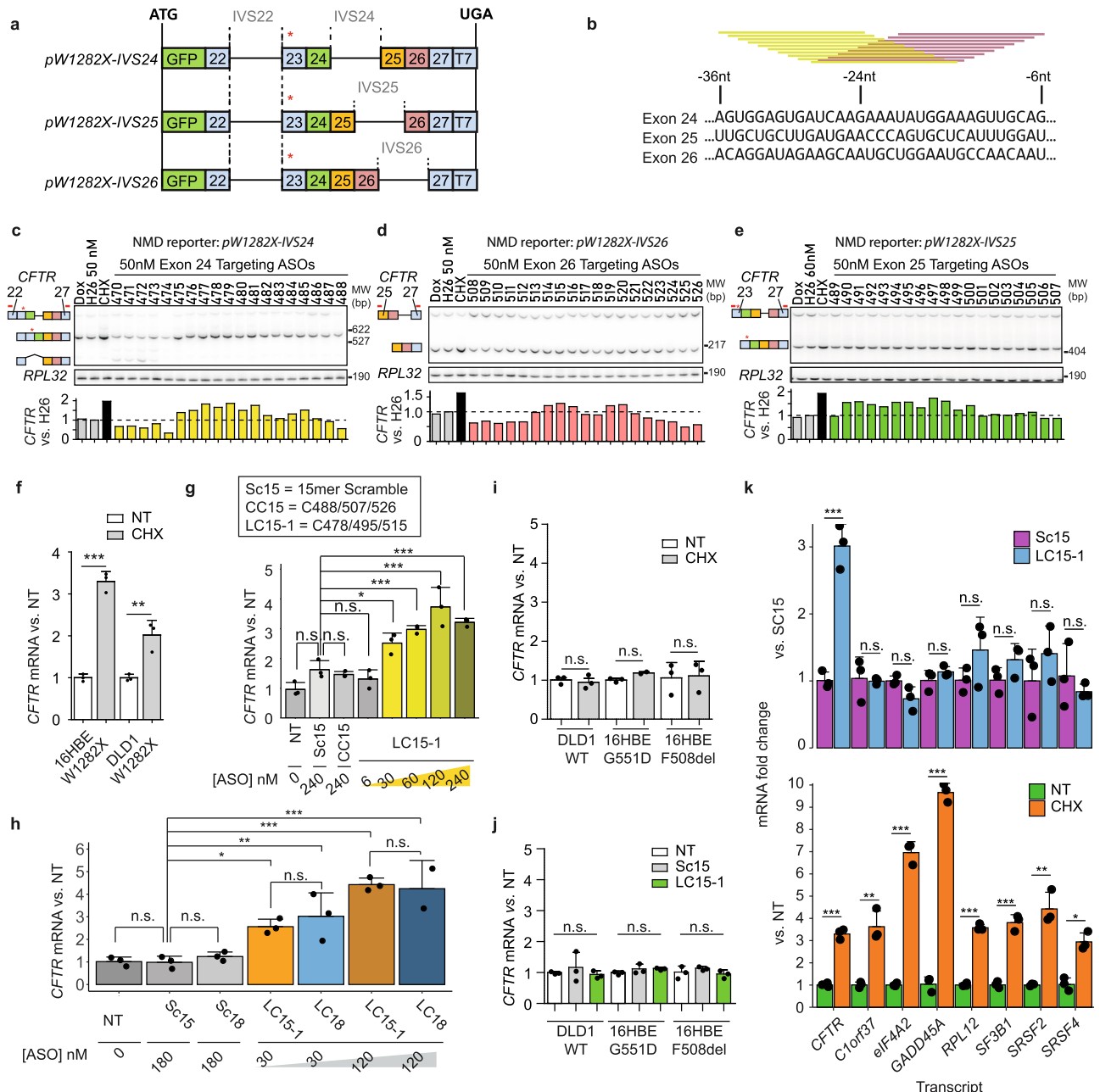

**Fig. 1 Identification of NMD-inhibiting ASOs and assessment of their specificity. a** Schematic of NMD reporters. The numbers show the *CFTR* exons in the NMD reporters. The red asterisk (*) indicates the location of the W1282X mutation. **b** Schematic of ASO screening. 19 MOE-PS-modified 15-mer ASOs (yellow and magenta bars) were designed to cover the presumptive EJC binding sites on exons 24, 25, and 26 at 1-nt resolution. U2OS cells stably expressing each NMD reporter were transfected with individual ASOs targeting EJC binding regions on *CFTR* exon (**c**) 24, (**d**) 25, or (**e**) 26, respectively. Reporter mRNA levels were measured by radioactive RT-PCR, using primers (red bars above the target exon) listed in Supplementary Table 5. **f** Effect of cycloheximide on *CFTR* expression in 16HBE-W1282X and DLD1-W1282X cells. **g** Effect of the 15-mer ASO cocktail LC15-1 on *CFTR* expression in 16HBE-W1282X cells. **h** Comparison between the effects of LC15-1 or LC18 on *CFTR* expression in 16HBE-W1282X cells. **i** Effects of cycloheximide on *CFTR* expression in DLD1-WT, 16HBE-G551D, and 16HBE-F508del cells. **j** *CFTR* mRNA levels in DLD1-WT, 16HBE-G551D, and 16HBE-F508del transfected with Sc15 or LC15-1 at a nominal total concentration of 120 nM. **k** Endogenous NMD-sensitive mRNA levels in 16HBE-W1282X cells treated with cycloheximide, 120 nM Sc15 or LC15-1. All mRNA levels in (**f**)–(**k**) were measured by RT-qPCR; *CFTR* mRNA levels were measured using forward and reverse primers targeting exon 22 and exon 23, respectively. *RPL32* served as internal reference for all panels except **h**, in which *HPRT* served as internal reference. NT = No treatment; Dox: doxycycline 1 μg/mL; Sc15/18 = 15/18-mer Scramble ASO; CC15 = ASO cocktail C488/C507/C526; LC15-1=ASO cocktail C478/C495/C515; LC18 = 18-mer ASO cocktail C24/25/26; CHX = 1-h incubation with 100 μg/mL cycloheximide. Data are represented as mean values ± SD. All data points represent independent biological replicates. **c**–**e** (*n* = 1). **f**–**k** (*n* = 3 for all treatments, except *n* = 2 in LC18-mer 120 nM in **h**). For all statistical tests, n.s. *P* > 0.05, *\*P* < 0.05, *\*\*P* < 0.01, *\*\*\*P* < 0.001. **f**, **h** (LC15-1 vs LC18), **i**, **k** two-tailed Student's *t*-test. **g**, **h** One-way ANOVA with Dunnett's post-test, versus Sc15. **j** One-way ANOVA. Source data are provided as a Source Data File.

We screened a total of 57 ASOs, uniformly modified with MOE ribose and a PS backbone. ASOs H24, H26, and M33 are uniformly MOE and PS-modified negative-control ASOs that are not complementary to any gene expressed in the reporter-expressing cells[28]. Based on the screen, we chose C478 and C515 as the initial lead ASOs targeting exons 24 and 26, respectively (Fig. 1c, d). Because the retention of IVS25 in *pW1282X-IVS25* caused by some ASOs prevented a clear assessment of NMD inhibition by the screened ASOs (Supplementary Fig. 2a), we generated a new *pW1282X-IVS25* NMD reporter with a stronger 5′ splice site (5′ss) but the same amino acid sequence (Supplementary Fig. 2b). Among several ASOs that increased the new NMD reporter levels, we chose C495 as the lead ASO (Fig. 1e). The candidate ASOs inhibited NMD of the reporters in a dose-dependent manner (Supplementary Fig. 2c–h).

**Identifying GAIN ASOs that inhibit NMD of *CFTR*-W1282X mRNA.** Endogenous *CFTR* mRNA is targeted for NMD in human bronchial epithelial cells and colon cancer cells harboring the homozygous *CFTR*-W1282X mutation (16HBE-W1282X and DLD1-W1282X cells) (Fig. 1f). The 16HBEge cell lines contain SV40 genomic sequence, which was used during the immortalization of the parental 16HBE14o- cell, within intron 6 of one *CFTR* allele[17]. As the SV40-containing allele does not express functional CFTR, the 16HBE-W1282X cells are functionally monoallelic[17]. We used two negative-control treatments: (i) a scrambled-sequence ASO based on C494 (Sc15); and (ii) a cocktail composed of C488/C507/C526 ASOs (CC15), which did not stabilize the NMD reporters. We first tested a lead 15-mer GAIN ASO cocktail composed of C478/C495/C515 (LC15-1), based on the above NMD-reporter screening results. Compared to the negative-control ASOs, LC15-1 significantly increased *CFTR* mRNA levels in both 16HBE-W1282X and DLD1-W1282X cells (Fig. 1g, Supplementary Fig. 3).

Length is an important parameter in ASO design that can affect the efficacy and specificity of uniformly modified ASOs[29,30]. Based on the results of the above 15-mer ASO screens, we designed a new 18-mer-ASO cocktail C24/C25/C26 (LC18) and an 18-mer scramble-ASO control (Sc18). Transfection of Sc18 did not increase *CFTR*-W1282X mRNA levels in 16HBE-W1282X cells, whereas LC18 increased *CFTR*-W1282X mRNA levels in a dose-dependent manner (Fig. 1h).

Based on the presumptive mechanism of action of the ASO cocktails, they should not affect the total mRNA levels of wild-type (WT) or missense-mutant *CFTR* mRNAs that are not sensitive to NMD (Fig. 1i). Indeed, *CFTR* mRNA levels were insensitive to transfection of the control ASO or LC15-1 in DLD1-WT, 16HBE-F508del, and 16HBE-G551D cells (Fig. 1j). Thus, LC15-1 only increased the mRNA levels of nonsense-mutant *CFTR*-W1282X mRNA. To test whether LC15-1 inhibits NMD of *CFTR*-W1282X mRNA specifically, as opposed to somehow affecting global NMD, we used RT-qPCR to survey seven other endogenous NMD-sensitive transcripts that are upregulated upon NMD inhibition by cycloheximide treatment[31,32]. As expected, LC15-1 increased *CFTR*-W1282X mRNA, without significantly changing any of the other NMD-sensitive mRNAs (Fig. 1k). LC18 also inhibited NMD of *CFTR*-W1282X mRNA in a gene-specific manner, as it did not affect two other endogenous NMD-sensitive mRNAs tested, *EIF4A2* and *SRSF2* (Supplementary Fig. 4).

To identify the optimal GAIN ASO cocktail, we performed a more comprehensive screening in 16HBE-W1282X cells (Fig. 2a–c). To systematically screen ASOs targeting each exon, we tested cocktails composed of two constant ASOs and one varying ASO. For example, to screen exon-24-targeting ASOs, we tested 19 cocktails composed

of varying exon-24-targeting ASOs and the same two ASOs targeting exons 25 and 26. After testing 57 such ASO cocktails, we identified C478/C494/C514 (LC15-2) as the new lead GAIN ASO cocktail, which was more potent than LC15-1 (Fig. 2d). At the highest concentration tested, both ASO cocktails increased *CFTR*-W1282X mRNA similarly, but at a lower concentration, the new lead cocktail had significantly higher potency.

**Mechanism of the GAIN lead ASO cocktail.** G542X and R1162X *CFTR* mutations are on exons 12 and 22, respectively. As these mutant mRNAs harbor more than three EJCs downstream of the premature termination codon, we tested whether their levels are insensitive to LC15-2. We transfected 16HBE-G542X and 16HBE-R1162X cells harboring homozygous G542X and R1162X mutations, respectively, with Sc15, CC15, or the lead ASO cocktail LC15-2 (Fig. 2e, f). Compared to the no-treatment control, transfection of 120 nM Sc15, CC15, or LC15-2 did not affect the levels of *CFTR*-G542X, *CFTR*-R1162X, and NMD-sensitive *EIF4A2* mRNAs. Only cycloheximide treatment caused significant increases in these NMD-sensitive mRNA levels. These results are consistent with the EJC-centric model of NMD, according to which at least one EJC > 55-nt downstream of a PTC is sufficient to induce strong NMD[20].

Using various combinations of the NMD-inhibiting ASOs, we next tested whether all three presumptive downstream EJC binding sites on *CFTR*-W1282X mRNA must be targeted with the corresponding ASOs for effective mRNA stabilization. Targeting only one or two EJC binding sites with the respective lead ASOs partially stabilized the *CFTR*-W1282X mRNA; on the other hand, the most significant and efficient increase in *CFTR*-W1282X mRNA was obtained by simultaneous transfection of all three lead ASOs with LC15-1, in both 16HBE-W1282X and DLD1-W1282X cells (Fig. 2g, Supplementary Fig. 5). Because ASO cocktails composed of one or two ASOs elicited a small increase in *CFTR*-W1282X mRNA levels, we next asked whether certain presumptive EJC binding sites may be more important than others. To test this possibility, we transfected 16HBE-W1282X cells with ASO cocktails with varying ratios of the individual ASOs (Fig. 2h). The LC15-1 cocktail with the highest total ASO concentration and an equimolar ratio of 40:40:40 nM increased *CFTR*-W1282X mRNA the most, and the increase was dependent on the total ASO concentration. In general, limiting the concentration of C495 in the cocktail reduced the *CFTR*-W1282X mRNA levels to the greatest extent (Supplementary Fig. 6a–c). Interestingly, ASO cocktails with equal total concentrations, but different ASO ratios, did not have equivalent effects on *CFTR*-W1282X mRNA levels. Also, some ASO cocktails increased *CFTR*-W1282X mRNA similarly or more than others, despite their lower total ASO concentration. The differences among *CFTR*-W1282X mRNA expression changes caused by the various ASO cocktails may be attributable to various factors, including partial EJC occupancy on different exons, differences in ASO uptake and target accessibility or affinity, involvement of EJC-independent NMD pathways, and RNA secondary structure[20,33].

**Splicing changes due to the GAIN ASO cocktails.** Some EJC-targeting ASOs may affect splicing, if their binding site overlaps with cis-elements that regulate splicing. We monitored exon 24–26 splicing by RT-PCR in DLD1-WT cells transfected with ASOs (Supplementary Fig. 7a). Exon-26-targeting ASOs did not detectably disrupt *CFTR* mRNA splicing. On the other hand, all exon-25-targeting ASOs caused slight exon-25 skipping, and some exon-24-targeting ASOs caused substantial exon-24

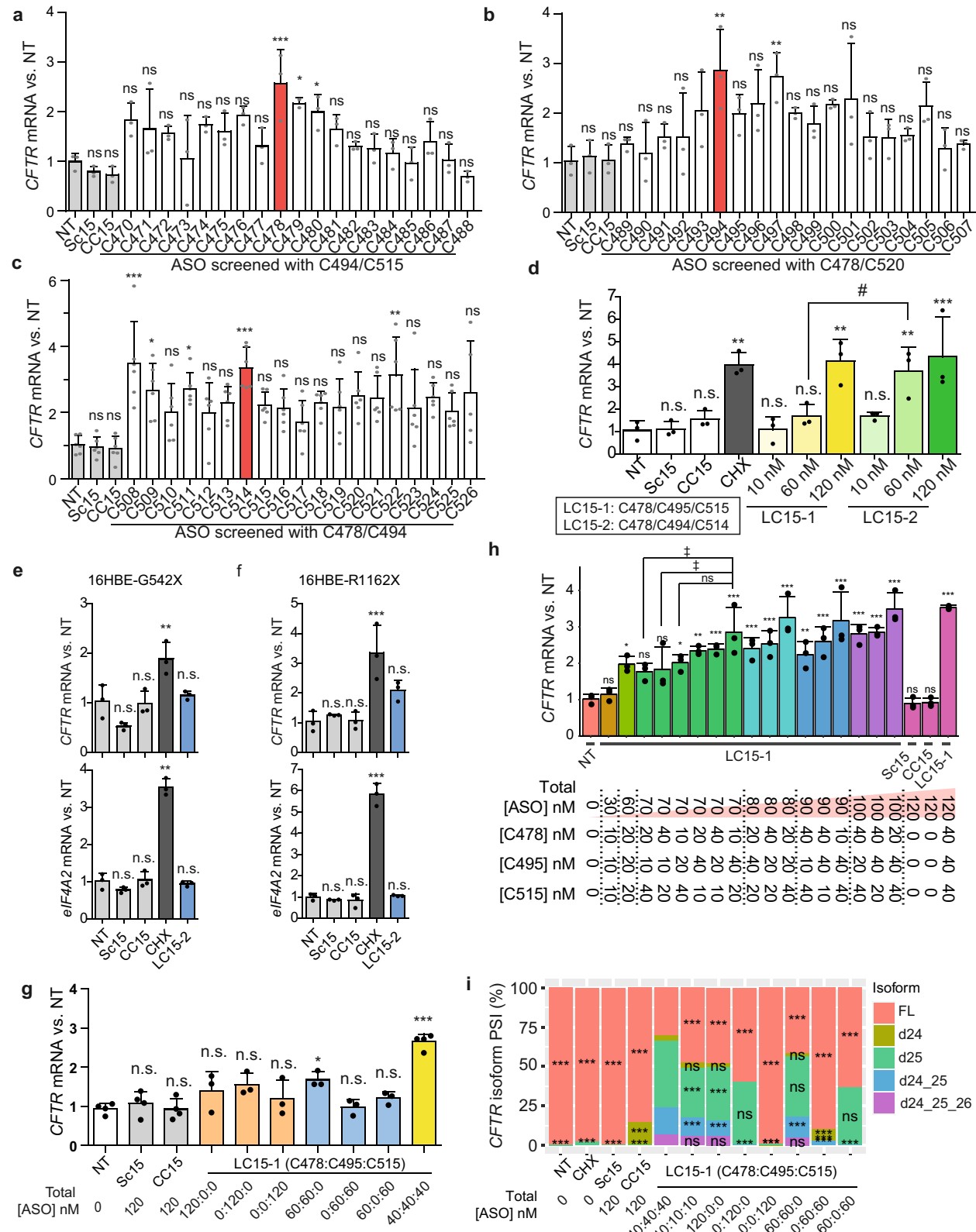

skipping. As disrupted binding of serine-rich (SR) proteins to exonic splicing enhancers (ESEs) by uniformly modified MOE-PS ASOs can cause exon skipping[34–37], we used ESEfinder[38] to predict putative ESEs on *CFTR* exons 24–26 that might be blocked by the lead ASOs (Supplementary Fig. 7b). mRNA sequences complementary to the lead ASOs C478 and C494 that caused exon 24 and 25 skipping overlap with SR protein motifs;

however, the target sites for the lead ASOs C514 and C515 that did not cause any exon 26 skipping overlapped with an SRSF5/SRp40 motif. Thus, overlap with an SR protein motif is insufficient to predict an ASO's interference with splicing.

Transfection of LC15-1 or LC18 caused dose-dependent, multiple exon skipping in human bronchial cells (Fig. 2i, Supplementary Fig. 8a–f): single (exon 24 or 25 skipping: d24

**Fig. 2 ASO cocktail optimization and mechanism of action.** ASOs targeting *CFTR* (**a**) exon 24, (**b**) exon 25, or (**c**) exon 26 were individually screened in 16HBE-W1282X cells, in combination with two ASOs that target the other two exons, at a total nominal concentration of 120 nM. Red bars indicate the lead ASO identified in each combination screen. **d** Comparison between LC15-1 and LC15-2. *CFTR* and *EIF4A2* mRNA levels in (**e**) 16HBE-G542X cells and (**f**) 16HBE-R1162X cells transfected with ASOs at a total nominal concentration of 120 nM. **g** The number of required EJCs targeted by ASOs C478, C495, C515, or all together, was assessed by transfecting 16HBE-W1282X cells with one, two, or three EJC-targeting ASOs at the same total nominal concentration. **h** *CFTR* mRNA in 16HBE-W1282X cells were transfected with various combinations of the lead ASOs C478, C495, and C515. **i**. Mean PSI of each *CFTR* isoform in 16HBE-W1282X cells transfected with various combinations of C478, C495, and C515. All mRNA levels in (**a**)–(**g**) were measured by RT-qPCR; *CFTR* mRNA levels were measured using forward and reverse primers targeting exon 22 and exon 23, respectively. *RPL32* mRNA level served as an internal reference. PSI measured in (**i**) used forward and reverse primers targeting exon 23 and exon 27, respectively. Abbreviations are as in Fig. 1; LC15-2 = C478/494/514. Data are represented as mean values ± SD. All data points represent independent biological replicates. **a**, **b**, and **d**–**f** ($n = 3$). **c** ($n = 6$ except $n = 5$ for 478/494/526). **h** ($n = 3$ except $n = 2$ for LC15-1 [40:40:40]). **g** ($n = 3$ except $n = 4$ for NT, Sc15, CC15, and LC15-1 [40:40:40]). **i** ($n = 4$ except $n = 3$ for LC15-1 [120:0:0], [0:120:0], [0:0:120], [60:60:0], [0:60:60], and [60:0:60]). For all Dunnett's post-tests, n.s. $P > 0.05$, */‡$P < 0.05$, **$P < 0.01$, ***$P < 0.001$. For all Student's *t*-tests, #$P < 0.05$. **a**–**h** One-way ANOVA with Dunnett's post-test versus NT, or Student's *t*-test. **h** ([total ASO] = 70 nM): one-way ANOVA with Dunnett's post-test vs LC15-1 [10:40:20]. **i** One-way ANOVA with Dunnett's post-test, versus each isoform in 'LC15-1 [40:40:40]'. Source data are provided as a Source Data File.

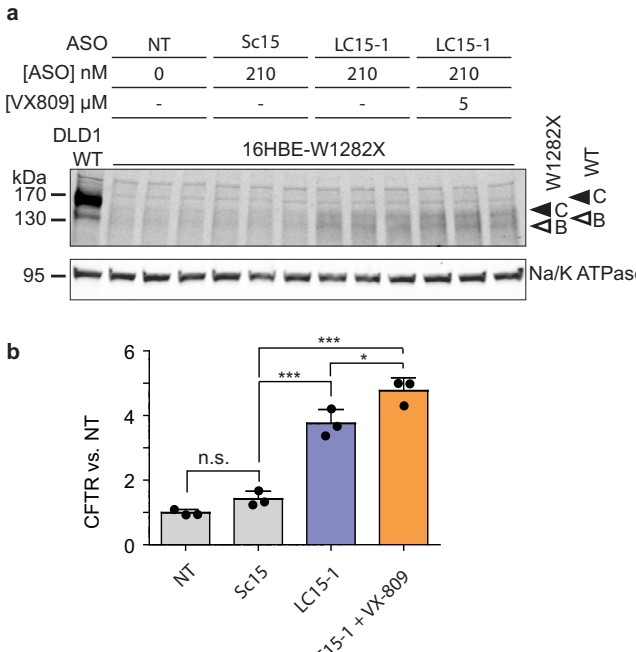

**Fig. 3 Effect of the lead ASO cocktail on CFTR-W1282X protein expression. a** Western blot of CFTR-W1282X protein in 16HBE-W1282X cells transfected with Sc15 or LC15-1 and treated with VX-809 at the indicated concentrations. The closed and open arrowheads indicate C and B bands of CFTR proteins, respectively. **b** Quantification of total CFTR protein in (**a**). Abbreviations as in Fig. 1. Data are represented as mean values ± SD. All data points represent independent biological replicates. $n = 3$, n.s. $P > 0.05$, *$P < 0.05$, **$P < 0.01$, ***$P < 0.001$, one-way ANOVA with Tukey's post-test. Source data are provided as a Source Data File.

and d25), double (exon 24 and 25 skipping: d24–25), and triple exon skipping (exon 24, 25, and 26 skipping: d24–25–26) of *CFTR* mRNA. Similar splicing changes occurred with LC15-1 or LC18 treatment by free uptake (Supplementary Fig. 9a, b). The splicing changes were not cell-line-specific, as the lead ASO cocktail promoted similar splicing changes in 16HBE-W1282X and DLD-W1282X cells (Supplementary Fig. 10a, b). The exon-skipping events occurred in the 3′UTR of the *CFTR*-W1282X mRNA, and thus should not affect the amino acid sequence of CFTR-W1282X protein. The length and sequence variation in the resulting 3′UTRs could have positive, negative, or no effect on translation efficiency.

CC15 caused a smaller degree of exon 24 and 25 skipping, consistent with the results in DLD1-WT cells (Fig. 2i). LC15-1 and LC18 generated the same *CFTR* isoforms, but with varying degrees of percent-spliced-in (PSI); for example, the PSI of the exon 24–25 double-skipping event was higher in 16HBE-W1282X cells treated with LC18 than LC15-1 (Supplementary Fig. 8c-d and 9b). Interestingly, whereas the 15-mer or 18-mer exon-26-targeting ASOs alone did not cause exon 26 skipping, the ASO cocktails containing exon-24-targeting ASO caused the appearance of the triple-skipped isoform (d24–25–26) (Fig. 2i, Supplementary Fig. 8c, d). To search for potential off-target sites on *CFTR* pre-mRNA, we looked for sites complementary to C478 with a maximum of four nucleotide mismatches, downstream of exon 24, and found only one site with four mismatches in intron 23. The chance of binding to an off-target with ≥ 4-nt mismatches is very low[39], suggesting that C478 is unlikely to cause exon 25 and 26 skipping by binding to ESEs in these exons. These results suggest that an ESE and/or the EJC in exon 24 is involved in long-range splicing regulation. Interestingly, recent studies showed that EJCs help maintain faithful splicing transcriptome-wide[40–46].

**The GAIN ASO cocktails increase CFTR protein**. Despite the splicing alterations, the increase in the total *CFTR* mRNA by the lead ASO cocktail resulted in increased CFTR-W1282X protein levels, compared to Sc15 (Fig. 3a, b). This result was expected, because none of the splicing changes affect the reading frame upstream of the nonsense mutation in exon 23. Combining the lead ASO cocktail with lumacaftor (VX-809), a corrector that improves the folding of CFTR protein[5,47], further increased the total CFTR-W1282X protein levels (Fig. 3a, b). Three different patterns of CFTR bands are visible on a Western blot: non-glycosylated A-band, core-glycosylated B band, and fully mature, glycosylated C-band[48]. As shown previously[5], truncated CFTR-W1282X exists as core-glycosylated and mature glycosylated forms (Supplementary Fig. 11a). Extensive enzymatic deglycosylation revealed that truncated-core and fully mature-glycosylated CFTR-W1282X proteins are upregulated by transfection of the C478/C494/C515 cocktail (LC15-3) in 16HBE-W1282X cells (Supplementary Fig. 11b, c).

**Intrinsic activity of recombinant CFTR-W1282X protein**. To understand the relative activity of CFTR-W1282X protein, compared to wild-type CFTR protein (CFTR-WT), we transduced 16HBE-W1282X cells with doxycycline (dox)-inducible wild-type CFTR-WT or CFTR-W1282X constructs using a lentiviral vector that allows co-expression of TurboGFP joined by a 2A self-cleaving peptide. In this way, we generated 16HBEge-GFP-P2A-

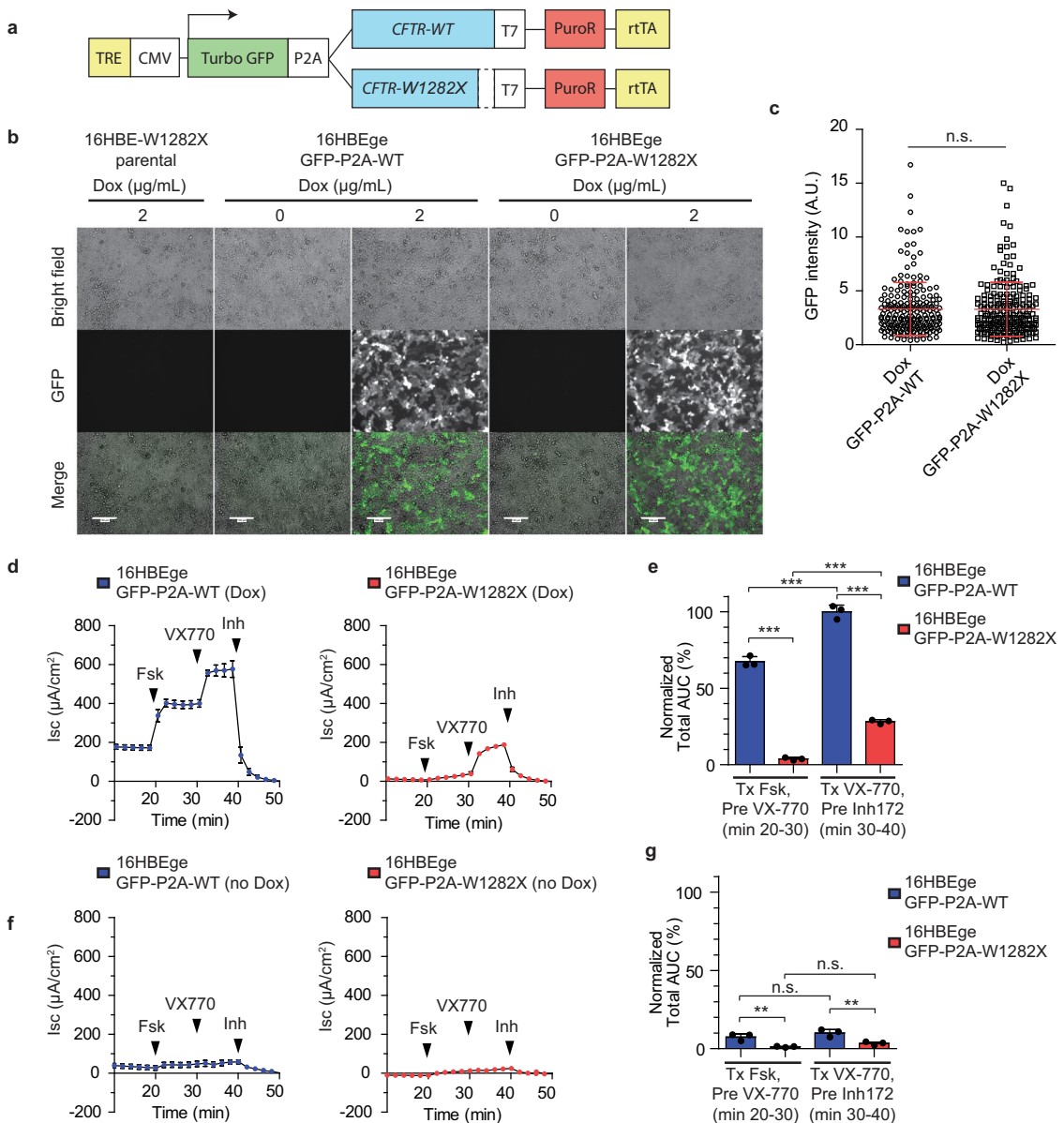

**Fig. 4 Intrinsic activity of CFTR-W1282X protein. a** Schematic of recombinant CFTR expression system. **b** Fluorescence microscopy images of 16HBE-W1282X, 16HBEge-GFP-P2A-WT, and 16HBEge-GFP-P2A-W1282X cells treated with doxycycline for 48 h. The white bars are 130-µm rulers. **c** TurboGFP signal intensity of 16HBEge-GFP-P2A-WT/W1282X cells treated with doxycycline (2 µg/mL) in (**b**) measured by fluorescence microscopy. **d** Average traces ($n = 3$) from Ussing-chamber assays of 16HBEge-GFP-P2A-WT/W1282X cells treated with doxycycline (2 µg/mL) and Trikafta. **e** The normalized total area under the curves of (**d**) between minutes 20–30 and 30–40. **f**. Average traces ($n = 3$) from Ussing-chamber assays of 16HBEge-GFP-P2A-WT/W1282X cells treated with Trikafta (VX-770/VX-661/VX-445) but no doxycycline. **g** The normalized total area under the curves of (f) between minutes 20–30 and 30–40. Data are represented as mean values ± SD. All data points represent independent biological replicates. **c** ($n = 210$ and $n = 208$ independent cells for 16HBEge-GFP-P2A-WT and 16HBEge-GFP-P2A-W1282X cells, respectively. n.s. $P > 0.05$, two-tailed Student's $t$-test). **d–g** ($n = 3$ n.s. $P > 0.05$, *$P < 0.05$, **$P < 0.01$, ***$P < 0.001$, one-way ANOVA with Tukey's post-test). Black arrowheads in (**d**) and (**f**) show when Fsk, VX-770, or CFTR inhibitor 172 was added; VX-661/445 treatment was started 24 h prior to the assay. TRE tetracycline-response-element, CMV CMV promoter, P2A self-cleaving 2A peptide, PuroR puromycin-resistance gene, rtTA reverse tetracycline transactivator, Dox doxycycline, Fsk 10 µM forskolin, AUC area under the curve, VX-770 10 µM VX-770, Inh 20 µM CFTR inhibitor 172. Source data are provided as a Source Data File.

WT and 16HBEge-GFP-P2A-W1282X cells (Fig. 4a). GFP levels served to control for expression from the integrated plasmid, as the TurboGFP and CFTR proteins are translated initially as a single polypeptide[49]. Fluorescence microscopy confirmed the induction of the recombinant gene expression by dox (Fig. 4b). GFP signals from 16HBEge-GFP-P2A-WT/W1282X cells treated with 2 µg/mL doxycycline were similar, indicating similar levels of induction of the recombinant CFTR proteins (Fig. 4c). The expression of recombinant CFTR-WT and CFTR-W1282X was

verified in the cells treated with or without doxycycline (Supplementary Fig. 12). The reduced level of CFTR-W1282X compared to CFTR-WT is attributable to the lack of C-terminal residues ([1478]TRL[1480]) important for post-translational processing and stability of CFTR protein[50,51].

Dox and Trikafta (VX-445, VX-661, and VX-770) treatment of 16HBEge-GFP-P2A-WT/W1282X cells led to a significant increase in CFTR activity, compared to Trikafta-only treatment, as quantified by the relative total area under the curve (AUC)

normalized to the total AUC after acute forskolin or VX-770 treatment (min 30–40) of 16HBEge-GFP-P2A-WT cells (Fig. 4d–g). The total AUC of 16HBEge-GFP-P2A-W1282X after VX-770 treatment was about 42 and 28% of the total AUC before and after VX-770 treatment in 16HBEge-GFP-P2A-WT cells, respectively (Fig. 4d, e). The background CFTR activity in 16HBE-W1282X cells due to endogenous CFTR-W1282X expression was negligible, as the chloride currents of the no-doxycycline-treatment 16HBEge-GFP-P2A-WT/W1282X cells were substantially lower than those of the doxycycline-treated cells (Fig. 4f, g).

**The effect of GAIN ASO cocktails on CFTR activity.** We next measured CFTR function in 16HBE-W1282X cells treated with the lead GAIN ASO cocktail, with the Ussing-chamber assay[52]. FDA-approved CFTR potentiators and correctors, such as iva-caftor (VX-770) and lumacaftor (VX-809), can enhance CFTR-W1282X activity in vitro, and may potentially benefit patients with the W1282X mutation[4–6,15]. Thus, we treated 16HBE-W1282X cells with CFTR modulators and the lead ASO cocktails by free uptake, and performed Ussing-chamber assays. The 16HBE-W1282X cells treated with LC15-2 and VX-770/809 increased CFTR-mediated chloride current, compared to 16HBE-W1282X cells treated with Sc15 and VX-770/809 treatment; LC18 and VX-770/809 treatment also significantly increased CFTR function, compared to the Sc18 and VX-770/809 treatment (Fig. 5a, b). We also tested the effect of lead ASO cocktail LC15-2 in cells co-treated with Trikafta (VX-770/661/445), which gave an even more significant CFTR activity enhancement (Fig. 5a, b). Consistent with the results from the ASO transfection experiments, LC15-2 or LC18 increased the *CFTR*-W1282X mRNA levels specifically, without affecting other endogenous NMD-targeted mRNAs in these cells (Supplementary Fig. 13a, b). This result demonstrates that gene-specific NMD inhibition of a hypomorphic *CFTR* allele leads to an increase in CFTR-mediated chloride current. We estimate that the combination treatment with the lead ASO cocktail and Trikafta results in CFTR-W1282X activity that is approximately 18 to 30% relative to the previously reported wild-type CFTR activity in 16HBE14o- cells[17]. Translation of all *CFTR*-W1282X mRNA isoforms generated by the ASO cocktail treatment presumably terminate at the W1282X codon, but the contribution of each isoform to CFTR activity may be affected by various factors, including mRNA stability, transport to the cytoplasm, and translational efficiency.

G418 is an aminoglycoside antibiotic that at high concentrations induces translational read-through of reporters containing *CFTR* nonsense mutations, and increases truncated CFTR-W1282X protein levels by NMD inhibition[17,53,54]. We observed that 200 μM G418 + VX-770/809 treatment or Sc18 + 200 μM G418 + VX-770/809 treatment increased CFTR activity in 16HBE-W1282X cells, compared to VX-770/809-only treatment, by further increasing the *CFTR* mRNA and CFTR protein levels in 16HBE-W1282X cells (Fig. 5a, b and Supplementary Fig. 14a, b). Full-length CFTR protein levels in 16HBE-W1282X cells reflecting read-through activity remained below the level of detection by Western blotting in our results and a previous report (Supplement Fig. 14a)[17]. Combining 200 μM or 600 μM G418 with LC18 + VX-770/809 further increased CFTR-W1282X function, compared to LC18 + VX-770/809 alone (Fig. 5a, b). To assess whether the increase in the CFTR activity is due to changes in the *CFTR* mRNA levels, *CFTR*-W1282X and endogenous NMD-targeted *SRSF2* mRNA levels were measured after all Ussing-chamber assays (Supplementary Fig. 13b). G418 inhibited NMD of *CFTR*-W1282X and NMD-targeted *SRSF2* mRNA in a dose-dependent manner. Compared to the *CFTR* mRNA levels in Sc18 + 200 μM G418 + VX-770/809 treatment, LC18 + 200 μM

G418 + VX-770/809 and LC18 + 600 μM G418 + VX-770/809 treatments led to higher *CFTR* mRNA levels, and these changes in the *CFTR* mRNA were sensitive to the G418 dose. These results show that the difference in CFTR activity between the lead LC18 + G418 + VX-770/809 treatment and Sc18 + G418 + VX-770/809 treatment is attributable to NMD inhibition by both the EJC-targeting ASOs and G418.

**Discussion**

NMD severely limits therapeutic development for CF caused by *CFTR*-W1282X mutation. Global NMD suppression can be achieved by targeting key NMD factors by gapmer ASOs that induce gene knockdown, or small molecules that inhibit the activity of NMD factors[16–19]. However, targeted NMD suppression may be more desirable, as it can avoid unwanted side-effects that may be caused by non-specific NMD inhibition. Here, we demonstrate that gene- and allele-specific NMD suppression using EJC-targeting ASO cocktails increases truncated CFTR-W1282X protein, as well as CFTR function. The combined treatment of the GAIN ASO cocktail and Trikafta in human bronchial epithelial cells with the W1282X mutation enhanced CFTR activity to 10–30% of wild-type CFTR. This result suggests that the combination treatment may enhance CFTR activity up to a therapeutic range. Consistent with this possibility, we estimate that the intrinsic activity of CFTR-W1282X protein, controlling for the plasmid expression levels, is approximately 28–42% of CFTR-WT protein, when combined with Trikafta.

The GAIN strategy we report here is specialized for the *CFTR*-W1282X mutation and may not be applicable for all *CFTR* mutation combinations, given the unexpected splicing changes caused by the lead ASO cocktail. Thus, our GAIN technology might be best suited for homozygous W1282X/W1282X mutations or compound heterozygous mutations with W1282X and a severe allele not amenable to currently approved therapies. For the GAIN technology to be applied to these compound hetero-zygous mutations, CFTR activity from the W1282X allele would have to increase to at least 20% of the wild-type CFTR activity to achieve phenotypic rescue. In the future, testing the GAIN ASO cocktails in a humanized mouse model with the W1282X mutation and patient-derived airway epithelial models could further help to evaluate their therapeutic potential.

Consistent with the EJC-centric model of NMD, our results demonstrate that the lead ASO cocktails inhibit NMD of *CFTR*-W1282X mRNA by preventing the binding of EJCs located downstream of the PTC, beyond the footprint of the stalled ribosome. The lead GAIN ASO cocktails inhibited NMD of *CFTR*-W1282X mRNA by targeting presumptive EJC binding sites downstream of the PTC, independently of cell type. They efficiently suppressed NMD of *CFTR*-W1282X mRNA only when all downstream EJC binding sites that contribute to NMD (i.e., those on exons 24–26) were targeted by the ASOs. On the other hand, the lead ASO cocktails did not affect the total mRNA levels of *CFTR* alleles without nonsense mutation (F508del and G551D) or *CFTR* alleles with nonsense mutations upstream of exon 23 (G542X and R1162X). The lead ASO cocktails did not inhibit NMD of other endogenous NMD-sensitive transcripts we tested, or affect the levels of NMD-insensitive *CFTR* mRNA. These observations rule out the possibility of global NMD suppression due to ASO treatment, or *CFTR* mRNA stabilization by inhibition of other mRNA-degradation pathways. These results also show that the GAIN ASO cocktails' effect inhibiting NMD, and thus increasing CFTR protein and chloride-transport activity, is specific for the W1282X allele, even though unintended exon-skipping occurred irrespective of the *CFTR* alleles. Based on the mechanism, a subset of the ASOs in the GAIN ASO cocktails

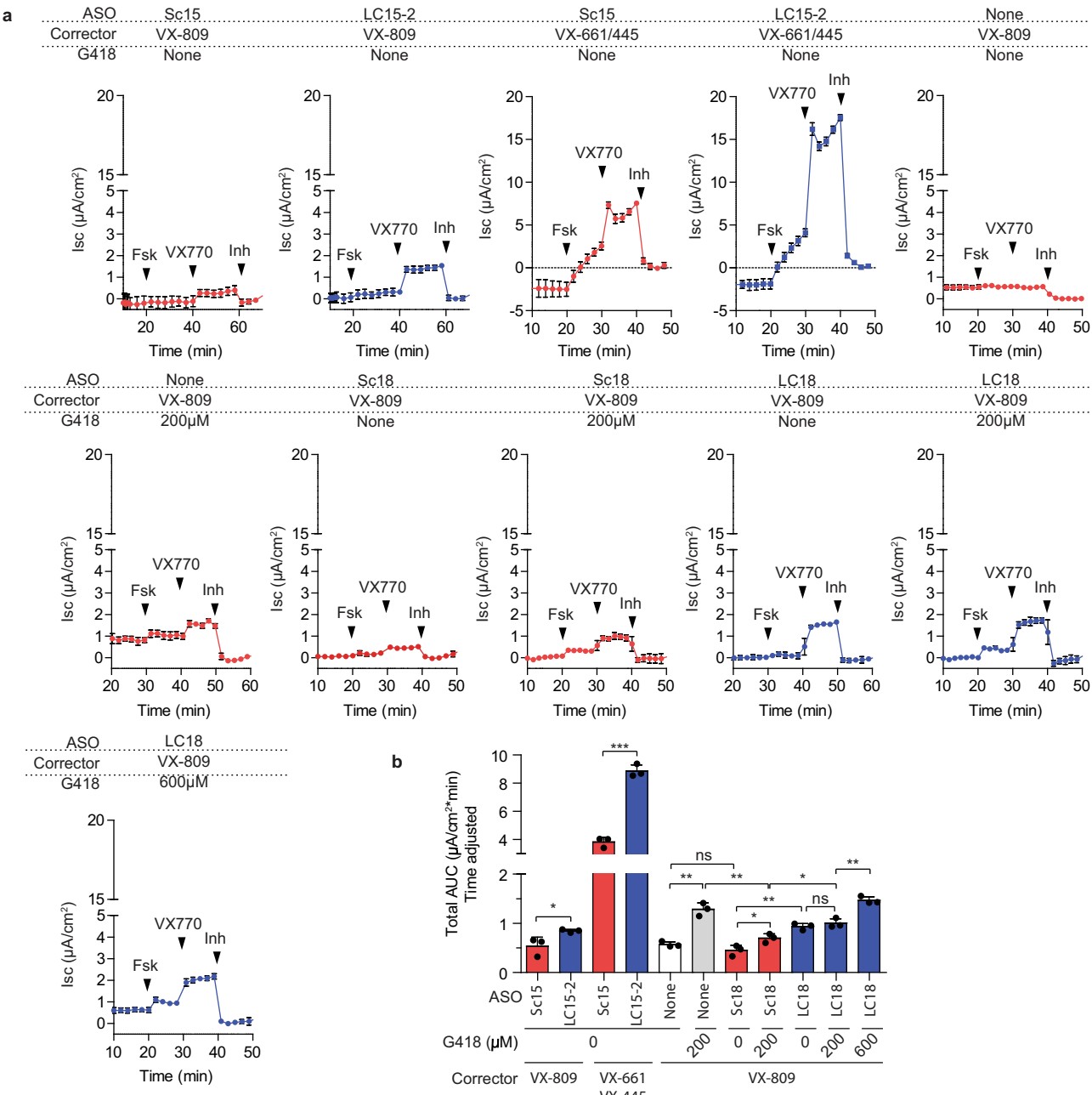

**Fig. 5 Effect of the lead ASO cocktail on CFTR-W1282X protein expression. a** Average traces ($n = 3$) from Ussing-chamber assay of 16HBE-W1282X cells treated with various combinations of 20 μM ASOs (Sc15, Sc18, LC15-2, or LC18), CFTR correctors (3 μM VX-809 or 3/3 μM VX-661/445), and G418 (200 or 600 μM), as indicated above each trace. All VX-809, VX-661/445, and G418 treatments were started 24 h prior to the assay. Black arrowheads show when Fsk, VX-770, or CFTR inhibitor 172 was added. **b** Total AUC in (a) divided by the total duration of each experiment (i.e., the time between forskolin treatment and CFTR inhibitor 172 treatment). Abbreviations as in Figs. 1 and 4. Data are represented as mean values ± SD. All data points represent independent biological replicates. $n = 3$ for all treatments. n.s. $P > 0.05$, *$P < 0.05$, **$P < 0.01$, ***$P < 0.001$, two-tailed Student's $t$-test. Source data are provided as a Source Data File.

might also be used to inhibit NMD of *CFTR* mRNAs with very rare nonsense mutations in exons 24 (e.g., Q1313X) or 25 (e.g., Q1330X and E1371X).

The lead 15-mer and 18-mer cocktails achieved similar levels of gene-specific NMD suppression, but had different effects on splicing. Compared to 12-mer ASOs, 18-mer ASOs tend to have fewer off-target effects on splicing[55], but whether our lead 18-mer ASO cocktail meaningfully reduces off-target effects and toxicity will require further investigation. We conclude that rationally designed ASOs can modulate clinically relevant

NMD, expanding the current RNA and oligonucleotide therapeutics toolbox.

The next stage in the development of the GAIN ASO cocktail as a potential therapy for CF caused by the W1282X mutation will require optimization of the delivery to the target tissues in in vivo models, such as mice. Systemic ASO treatment is suboptimal for delivering ASOs to the lungs and airways[56]. In contrast, aerosolized ASO is a more clinically relevant method of intratracheal delivery that may maintain sufficient tissue concentration in the airways[57]. In mice, the bioavailability of

aerosolized MOE ASO is 1260% compared to ASO administered intravenously[56]. In the lungs of mice and monkeys, the aerosolized ASOs have half-lives of 4 days and more than 7 days, respectively[56]. Aerosolized gapmer ASOs can efficiently knockdown target mRNAs and have low toxicity, low systemic exposure, and good lung-tissue accumulation (i.e., 0.1–0.3 mg/kg of aerosolized ASO resulting in 10 µg/g lung-tissue concentration)[58].

Several clinical trials have already shown that aerosolized ASOs can be efficiently delivered to the airway tissues, although further studies to demonstrate safety and efficacy will be essential. Topigen developed an inhaled ASO cocktail, TPI ASM8, composed of two PS ASOs, TOP004 and TOP005, targeting CCR-3, IL-3, IL-5, and GMCSF for the treatment of mild asthma[59]. TPI ASM8 knocked down its target genes efficiently in rodents and non-human primates[60]. Phase-2 clinical trials (NCT01158898, NCT00264966, and NCT00822861) demonstrated the safety of TPI AMS8 in humans. The reduction in late-phase allergen response (primary outcome measure) between TPI ASM8 and placebo was statistically significant in the open-label study (NCT00822861), but not in a double-blinded placebo-controlled study (NCT01158898)[61]; however, TPI ASM8 reduced inflammatory response and sputum eosinophils in dose-escalation studies[60]. Ionis Pharmaceuticals developed an aerosolized gapmer ASO, IONIS-ENaC-2.5-Rx, targeting *SCNNA1*. It showed significant in vivo activity and no safety concerns in a phase-1/2a clinical trial (NCT03647228); however, due to long-term toxicity observed in the preclinical model, its development was recently discontinued[62]. QR-010 is a uniformly 2′O-methyl- and PS-modified ASO developed by ProQR Therapeutics that converts *CFTR*-F508del mRNA into WT *CFTR* mRNA by inserting the three missing bases in the mutant through an unknown mechanism, and is currently in a phase-1b clinical trial[63]. There are still various barriers for developing aerosolized ASOs; the properties of the airway (e.g., geometry, humidity, drug clearance, and pathology) and the aerosolized ASO droplets (e.g., size, charge, etc.,) can all affect the efficacy and toxicity of ASOs[64]. However, these ongoing studies illustrate the potential of aerosolized nucleic-acid therapeutics for CF and other lung diseases.

## Methods

**ASOs.** All ASOs were uniformly modified with 2′-*O*-(2-methoxyethyl) (MOE) ribose, phosphorothioate (PS) linkages, and 5′methylcytosine. The 15-mer ASOs were obtained from Ionis Pharmaceuticals (Carlsbad, CA) and Integrated DNA Technologies (Coralville, IA), and 18-mer ASOs were obtained from Bio-Synthesis (Lewisville, TX). All ASOs were dissolved in water and stored at −20 °C. Stock ASO concentrations were calculated based on the A260 measurement and each ASO's extinction coefficient $e$ (mM$^{-1}$ × cm$^{-1}$ @ 260 nm). The sequences of all ASOs used in this study are listed in Supplementary Table 1; the list of control and lead ASO cocktails is provided in Supplementary Table 2.

**Preparation of U2OS cells expressing NMD reporters.** The NMD reporters used for the ASO screening (*pW1282X-IVS23, pW1282X-IVS24, pW1282X-IVS25,* and *pW1282X-IVS26*) were constructed from the parent NMD reporter *GFP-CFTR22-27-T7*, which was cloned into the pCDNA5 FRT/TO plasmid (Life Technologies, Carlsbad, CA). pCDNA5 FRT/TO allows tetracycline-inducible expression of the gene. *GFP-CFTR22-27-T7* has the natural sequences of exons 22–27 of the human *CFTR* gene and shortened intervening sequences (IVS) modified from the natural sequences of introns 22–26 by taking 200 nucleotides (nt) from the 5′ and 3′ ends of the corresponding introns. GFP and T7 cDNA sequences were added to the 5′ and 3′ ends of each reporter, respectively, to facilitate gene, transcript, and protein detection. NMD reporters *pW1282X-IVS23, pW1282X-IVS24, pW1282X-IVS25,* and *pW1282X-IVS26* comprise only IVS23, IVS24, IVS25, or IVS26, respectively, downstream of the PTC. All NMD reporters have IVS22 upstream of the exon 23 sequence. The 5′ss and 3′ss of IVS25 in *pW1282X-IVS25* were mutated to stronger splice-site sequences (Supplementary Fig. 2B) to promote proper splicing in the minigene context.

For stable expression of NMD reporters, the reporter plasmids were co-transfected with the pOG44 helper vector to express Flp recombinase into U2OS-TREx cells harboring a single FRT recombination site (Life Technologies). Cells with successful NMD-reporter integration were selected by hygromycin resistance. The expression and splicing of the transgenes were assessed by radioactive RT-

PCR, following induction with 1 µg/ml doxycycline (Research Products International Corp, D43020-100).

**CRISPR mutant DLD1 and 16HBEge cells.** Using CRISPR/Cas9, we generated DLD1 cells with homozygous *CFTR*-W1282X mutation. sgRNA against exon 23 (Supplementary Table 3) was cloned downstream of the U6 promoter of the pSpCas9(BB)-2A-GFP (PX458) plasmid (Addgene plasmid # 48138)[65], creating pSC2G-CFTR23, which allows co-expression of the sgRNA and *Streptococcus pyogenes* Cas9 (spCas9). Four micrograms of pSC2G-CFTR23 plasmid was co-transfected with 1 µM single-stranded DNA repair template (synthesized by Sigma, Supplementary Table 3) comprising the *CFTR*-W1282X and silent protospacer adjacent motif (PAM) mutations, using lipofectamine 2000 (Life Technologies, 11668019) according to the manufacturer's protocol. Following transfection, GFP + DLD1 cells were collected using an ARIA-I cell sorter (BD), and individual clones of cells were isolated by limiting dilution in 96-well plates. Clonal cells were expanded and passaged until confluent in 6-well plates. We characterized 159 clones by Sanger sequencing, and identified two heterozygous and two homozygous W1282X mutant clones. 16HBE14o- parental cells gene-edited to yield 16HBEge cell lines CFF-16HBEge CFTR-W1282X, F508del, G551D, G542X, or R1162X, homozygous for *CFTR*-W1282X, F508del, G551D, G542X, or R1162X mutation, respectively, in the endogenous loci were kindly provided by the Cystic Fibrosis Foundation Therapeutics (CFFT) Lab[17]. Elsewhere in the text, these cells are referred to as 16HBE-W1282X, 16HBE-F508del, 16HBE-G551D, 16HBE-G542X, or 16HBE-R1162X cells, respectively.

**Tissue culture and transfection of siRNA, ASO, and plasmids.** U2OS and DLD1 cells were cultured in DMEM with 10% FBS. 16HBEge cells were cultured in MEM with 10% FBS. All cells were incubated at 37 °C and 5% CO$_2$. Cells were transfected with ASOs and plasmids using Lipofectamine 3000 (Life Technologies, L3000015) according to the manufacturer's protocol, and harvested 48 h post transfection. Cells were transfected with siRNA (Supplementary Table 4) using Lipofectamine RNAiMax (Life Technologies, 13778075) according to the manufacturer's protocol for transfecting short oligonucleotides, and harvested 48 h post transfection. NMD inhibition by cycloheximide was performed by treating the cells for 1 h with cycloheximide (Sigma, 100 µg/mL). For ASO treatment by free uptake, 1 mM stock ASO solutions were diluted into MEM with 10% FBS to the desired final concentrations, and the cells were cultured for 4 days before harvesting or Ussing-chamber assays. NMD-reporter expression was induced with 1 µg/ml doxycycline with media change, 6 h after transfection. For the G418-treatment group, G418 (Sigma) was added to the culture medium at the indicated final concentrations, 24 h before protein extraction or Ussing-chamber assays.

**CFTR expression plasmid.** The plasmid pcDNA5.FRT-wtCFTR comprising the coding sequence for the WT *CFTR* gene was a kind gift from the CFFT Lab. The plasmid contains the *CFTR* gene coding sequence in the pCDNA5 FRT/TO plasmid backbone (Life Technologies). The *CFTR* coding sequence from pcDNA5.FRT-wtCFTR tagged with T7 at the carboxyl-terminus was cloned between the AvrII and BsrGI multiple cloning sites of the pCW57-GFP-2A-MCS plasmid; the resulting plasmid was dubbed pGFP-P2A-CFTR-WT-T7. The pCW57-GFP-2A-MCS plasmid was a gift from Adam Karpf (Addgene plasmid No. 71783; n2t.net/addgene:71783; RRID: Addgene_71783). We introduced the PTC at amino acid position 1282 (*W1282X*) into pGFP-P2A-CFTR-WT-T7 to generate the pGFP-P2A-CFTR-W1282X-T7 plasmid.

**Generation of 16HBEge-GFP-P2A-WT/W1282X cells.** *CFTR*-containing lentiviral plasmids (pGFP-P2A-CFTR-WT-T7 and pGFP-P2A-CFTR-W1282X-T7) and packaging vector were transfected into HEK293T cells for producing retroviral particles. All plasmids were transfected using Lipofectamine 2000 (Life Technologies, 11668019) according to the manufacturer's protocol. Fresh medium was added to the cells after incubation for 18 h after transfection. The medium containing the viral particles was collected 24 h later and replaced with fresh medium. The viral-particle-containing medium was collected again after 24 h and pooled with the previous supernatant. The pooled medium was filtered through a 0.45 µm filter. PEG-it™ solution (System Biosciences, LV810A-1) was diluted into the filtered medium to a final 1/5 dilution, and the diluted mixture was incubated at 4 °C for 6 h. After centrifugation for 30 min at 1500 × *g*, the supernatant was added to the 16HBE-W1282X cells. The transduced cells were selected with puromycin (0.2 µg/ml) (EMD Millipore, 540411) for one week, starting 18 h after the addition of viral particles.

**Fluorescence microscopy.** 16HBEge-GFP-P2A-WT/W1282X cells were incubated in growth medium containing 2 µg/mL doxycycline (Research Products International Corp, D43020-100) for 48 h to induce the expression of recombinant TurboGFP-P2A-CFTR proteins. A Revolve microscope was used for imaging, and the images were acquired by ECHO Pro v6.0.1 software (Revolve, ECHO, San Diego, CA, USA). GFP-intensity analysis from the images was performed with ImageJ2 software (version 2.2.0)[66].

**RNA extraction and RT-PCR.** Total RNA was extracted with TRIzol (Life Technologies) according to the manufacturer's protocol. Oligo dT(18)-primed reverse transcription was carried out with ImProm-II Reverse Transcriptase (Roche). Semi-quantitative radioactive PCR (RT-PCR) was carried out in the presence of $^{32}$P-dCTP with AmpliTaq DNA polymerase (Thermo Fisher), and real-time quantitative RT-PCR (RT-qPCR) was performed with Power Sybr Green Master Mix (Thermo Fisher). Primers used for RT-PCR and RT-qPCR are listed in Supplementary Table 5. RT-PCR products were separated by 6% native polyacrylamide gel electrophoresis, detected with a Typhoon FLA7000 phosphorimager, and quantitated using MultiGauge v2.3 software (Fujifilm); RT-qPCR data were quantitated using QuantStudio 6 Flex system.

**Protein extraction, deglycosylation, and western blotting.** Cells were harvested with RIPA buffer (150 mM NaCl, 50 mM Tris-HCl pH 8.0, 1% NP40, 0.5% sodium deoxycholate, and 0.1% SDS) and 2 mM EDTA + protease inhibitor cocktail (Roche) by sonicating for 5 min at medium power using a Bioruptor (Diagenode), followed by 15-min incubation on ice. Protein concentration was measured using the Bradford assay (Bio-Rad) with BSA as a standard. To monitor post-translational maturation of CFTR protein, cell lysates were incubated in 40 µg/ml PNGase F (New England Biolabs, MA) for 2 h at 37 °C to cleave all N-glycans before immunoblotting[67]. Cell lysates were mixed with Laemmli buffer and incubated at 37 °C for 30 min. The protein extracts were separated by sodium dodecyl sulfate-polyacrylamide gel electrophoresis (SDS-PAGE) (6% Tris-chloride gels) and then transferred onto a nitrocellulose membrane. CFTR bands C, B, and A were detected with antibody UNC-596 (J. Riordan lab, University of North Carolina, Chapel Hill, NC). The C and B band intensities were measured together for the quantification of CFTR protein levels. The specificity of the antibody was confirmed by knocking down CFTR in WT DLD1 cells (Supplementary Fig. 11A). UPF1 was detected with rabbit antibody D15G6 (Cell Signaling Technology #12040S). Na/K-ATPase, detected with a mouse monoclonal antibody (Santa Cruz, sc-48345), and Tubulin, detected with rabbit antibody (GenScript, Cat# A01203), were used as loading controls. IRDye 800CW or 700CW secondary antibody (LI-COR) was used for Western blotting, and the blots were imaged and quantified using an Odyssey Infrared Imaging System (LI-COR). Statistical significance was calculated using Student's t-test or one-way ANOVA, followed by Tukey's or Dunnett's post-test.

**Ussing-chamber assay**

*Preparing 16HBEge cells for Ussing-chamber assay.* 16HBEge cells were grown as an electrically tight monolayer on Snapwell filter supports (Corning, cat# 3801), as described[68]. In brief, the 16HBEge cells were seeded on Snapwell filter supports at a density of $5\times10^5$ cells/well. The Snapwell filter inserts were pre-coated with human placental collagen type VI (Sigma, C5533-5MG). Both serosal and mucosal membranes were exposed to the ASOs for 4 days, and to CFTR correctors for 24 h, before the assays. The cells were cultured for a total of 6–7 days before the Ussing-chamber assay. The Snapwell inserts were transferred to an Ussing-chamber (P2302, Physiologic Instruments, Inc., San Diego, CA). For 16HBEge cells, the serosal side only was superfused with 5 mL of HEPES-buffered physiological saline (HB-PS) buffer; on the mucosal side, 5 ml of CF-HEPES-buffered physiological saline (CF-PS) was used (137 mM Na-gluconate; 4 mM KCl; 1.8 mM CaCl$_2$; 1 mM MgCl$_2$; 10 mM HEPES; 10 mM glucose; pH adjusted to 7.4 with N-methyl-D-glucamine) to create a transepithelial chloride-ion gradient. After clamping transepithelial voltage to 0 mV, the short-circuit current ($I_{SC}$) was measured with a Physiologic Instruments VCC MC6 epithelial voltage clamp, while maintaining the buffer temperature at 37 °C. Baseline activity was recorded for 20 min before agonists (final concentrations: 10 µM forskolin (Sigma, F6886)), and 1–10 µM VX-770 (Selleckchem, S1144) and inhibitor (final concentration: 20 µM CFTRinh-172 (Sigma, C2992)) were applied sequentially at 10 or 20-min intervals, to both serosal and mucosal surfaces. Agonists/inhibitor were added from 200x-1000x stock solutions. Data acquisition was performed using ACQUIRE & ANALYZE Revision II (Physiologic Instruments).

**ESE motif analysis.** Potential SR protein binding sites were analyzed by ESEfinder[38].

**Statistical analyses.** Statistical analyses were performed with GraphPad Prism 5. Statistical parameters are indicated in the figures and legends. For two-tailed Student's t-test or one-way analysis of variance (ANOVA) with Tukey's or Dunnett's post-test, $P < 0.05$ was considered significant. The asterisks and hash signs mark statistical significance as follows: n.s. $P > 0.05$; $^{*/\#}P < 0.05$; $^{**/\#\#}P < 0.01$; $^{***/\#\#\#}P < 0.001$. All exact P values are provided in the Supplementary Data 1.

**Reporting summary.** Further information on research design is available in the Nature Research Reporting Summary linked to this article.

## Data availability

The data supporting the findings of this study are available from the corresponding authors upon reasonable request. Source data for the figures and supplementary figures are provided as a Source Data file.

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

## Acknowledgements

We are very grateful to Martin Mense and Hermann Bihler (The CFFT Lab, Lexington, MA) for generously sharing protocols and advice, and for helpful comments on the manuscript. This work was supported by Cystic Fibrosis Foundation grant KRAINE17GO and NIH grant R37GM42699 to A.R.K. Y.J.K. was supported by NIH grants F30HL137326-04 and T32GM008444. We acknowledge assistance from Cold Spring Harbor Laboratory Shared Resources, funded in part by NCI Cancer Center Support Grant 5P30CA045508.

## Author contributions

Y.J.K., T.N., and A.R.K. conceived the study. A.R.K. supervised the study. Y.J.K. and T.N. generated the DLD1-W1282X cells. F.P. performed exon 25-targeting ASO screening using the *pW1282X-IVS25* NMD reporter. L.Y. and Q.Z. generated lentiviral 16HBEge cell lines. Y.J.K. designed and performed all other experiments and analyzed the data. Y.J.K. and A.R.K. wrote the paper, and all authors approved the manuscript.

## Competing interests

The authors declare the following competing interests: A.R.K. and T.N. are inventors in issued patent US20160194630A1, "Reducing nonsense-mediated mRNA decay", assigned to Cold Spring Harbor Laboratory. The authors Y.J.K., F.P., L.Y., and Q.Z. declare no competing interests.
