## [Peer Review File · Nature Communications]

Title: Gene-Specific Nonsense-Mediated mRNA Decay Targeting for Cystic Fibrosis TherapyREVIEWER COMMENTS

Reviewer #1 (Remarks to the Author):

The manuscript from Kim et al. describes a method for increasing the level of expression of a truncated but still useful protein expressed by CFTR genes carrying the W1282X mutation. The level of expression is low because the termination codon in the mutant is upstream of several introns, and splicing of these introns leads to the deposition of exon junction complexes and thence to nonsense-mediated decay (NMD) of the mRNA. Kim et al. demonstrate that a cocktail of modified oligonucleotides can be selected that blocks NMD and increases expression of the truncated protein, with functional benefits in a cell line carrying the mutation. The results are significant for three reasons: first, they open up the prospect of a therapy for cystic fibrosis patients carrying the mutation, second, they demonstrate the feasibility of suppressing NMD at a number of splice junctions and, third, the strategy used to optimise each oligonucleotide before combining them is an excellent example to others seeking to achieve similar results. The results will stimulate investigations in other genes where the premature termination codon is upstream of several introns.

Figure 1 shows the initial optimization of the oligonucleotides, in which minigenes were used that carry the mutation and only one of the three introns being examined. This is followed by experiments using a cocktail of three oligonucleotides, the optimal one for each intron, in cell lines carrying the homozygous mutation, and then the specificity of the response was analysed using a panel of genes shown by cycloheximide treatment to be limited by NMD. Figure 2 shows a repeat of the screen in which one oligonucleotide was varied while the other two optimal oligonucleotides were held constant, followed by an analysis of the optimal ratios of the oligonucleotides. It also shows the effects of the oligonucleotides on splicing, since they would be expected to inhibit splicing of the proximal intron or induce skipping of the targeted exon. In practice, the oligonucleotides caused a complex pattern of exon skipping but this did not appear to prevent some increase in protein levels (Figure 3), possibly because the skipped exons were downstream of the PTC. The effects on splicing were only detected because the authors used radioactive PCR as well as qPCR.

In general, the work was done to a very high standard and the systematic exploration of sequence, length, combinations and outcomes was impressive. I have only minor suggestions for improvements.

1. The figure legends should state the exons being targeted by the PCR primers. This is important when comparing the outcomes of splicing analysis (Figure 2I, and Supp. Figs 7-10) with the measurements of mRNA made in Figure 2 (A-G) and Supp. Figs 2-6.

2. There is an unfortunate disjunction in the oligonucleotides used. At the foot of page 5 it is asserted that the lead cocktail contained C495 (with C478 and C515). However, the following line refers to Supp. Fig 2 where, in panels D and G, C496 was used. Later the lead cocktail became C478+C494+C514 (Figure 2D, E and F), but Figs 2G and H, using the lead ASOs, turned out to have used C495+C515 (also in Supp. Figs 5, 6 and 8). Supp. Figs 9 and 11 used C494+C515. In all these cases the mixture is described as the

lead cocktail. Figure 3 used C494+C515 (A) and C494+C514 (D). It is unlikely that the changes mattered, in terms of outcomes, but it would be helpful if the authors could find a way to either state that or indicate in the main text which 'lead cocktail' is being used.

3. Supplementary Figure 3 has 17 bars and 18 labels.

4. Panels C and D in Supp. Fig. 11 are mislabeled.

5. The ability of G418 to increase expression of truncated but not full-length CFTR needs some discussion. Apart from an undetectable increase in full-length protein, is there any other reasonable explanation of its functional effects?

Reviewer #2 (Remarks to the Author):

This study is designed and performed convincingly. Especially, the authors put in a significant amount of work to set up a system to screen ASOs: (1) The authors established a human bronchial epithelial cell line with homozygous CFTR W1282X mutations by CRISPR knock-in. This cell line allowed for testing of the functional rescue (CFTR-mediated chloride current) by ASO cocktails, (2) The authors developed three minigene-based reporters to screen ASOs that, each, target individual EJs. Based on this system, the authors found ASO cocktails that significantly increase CFTR mRNA and protein levels as well as CFTR-mediated chloride current when combined with VX770, VX809, and/or G418.

While the study has many limitations (listed below), it is nonetheless a high-quality study demonstrating the potential of NMD-inhibition ASO approach for CF treatment. With earnest revision efforts to mitigate some of the addressable limitations, this manuscript is considered to have enough merits to be published in Nature Communications.

Limitations:

- Novelty and innovativeness: As mentioned in the manuscript (line 81), the authors had already published the concept of ASO-mediated NMD inhibition. Hence, the novelty and innovativeness of the current study are limited to the application of the concept to CF.

- Clinical applicability - delivery issues: Cystic fibrosis is a disorder that mainly affects the lung. Therefore, deliverability to the lung, precisely to bronchial epithelial cells, is one of the most critical factors for successful CF drug development. Although multiple preclinical studies have demonstrated that ASOs can be delivered to the lung as aerosols, it has not been proven clinically (no FDA approved ASO drugs targeting the lung). Hence, the lung delivery issue remains a significant hurdle for the clinical application of this study.

- Clinical applicability - issues related to cocktail treatment: The authors showed that the lead ASO cocktail is effective when all three ASOs are administered at the same time and with a specific stoichiometry (1:1:1 ratio). This requirement poses two major challenges for the clinical application. The first challenge pertains to efficacy. It is not given that the three ASOs have equal in vivo distribution and cellular uptake rates. The potential difference in the distribution/uptake rates may lead to skewed stoichiometry deviating from the optimal ratio and, hence, diminished efficacy. The second challenge pertains to tolerability. A cocktail of three ASOs comes with three times the risk of unexpected toxicity as monotherapy with a single ASO. Even if the authors demonstrate that the three ASOs do not have critical off-targets (by sequence alignments or RNA-seq), it does not preclude that at least one of three ASOs have some unpredictable toxicity. The higher risk of toxicity may hamper the clinical application of the study.

- Functional evidence: It's not clear if the extent of CFTR functional restoration shown in Fig 3 is enough to produce symptomatic benefit in patients. The authors did not mention how much increase in the CFTR-mediated chloride current should be enough for symptomatic benefit. The authors did mention that "As low as 10% of normal CFTR function provides a significant therapeutic benefit for CF patients", but this "10%" refers to the gene expression level, not the chloride current level. Also, the authors did not mention whether or not the observed increase in the chloride current by the ASO cocktail corresponds to more than 10% of the normal current level (the normal current level, an essential reference, is not provided in this study). Judging from the fact that the ASO cocktail led to only a low level of increase of the partially functional (with skipped exons and readthrough) protein (Fig. 3A), the level of functional restoration could be lower than 10%.

- Potential side effect: The authors showed that the ASO cocktail leads to undesirable exon skipping events. These exon skipping events can adversely impact the function of CFTR protein, especially when a patient treated with the cocktail is compound heterozygous for the W1282X mutation. Since the ASO cocktail can further damage the function of the other allele by inducing exon skipping, the allele might not be able to function properly even with the help of other CF drugs, such as VX770, VX809, and/or G418. Thus, it would be helpful to demonstrate that the ASO cocktail does not diminish the CFTR function in bronchial epithelial cells of normal individuals and CF patients with non-W1282X mutations.

- ESE prediction: The authors used ESEfinder to identify ESEs, only to dismiss the ESE prediction result as inconclusive because it is only a prediction (Line 203-205). Perplexingly, the authors then reference the prediction results to interpret some of the exon skipping events. The authors may consider clarifying these seemingly self-contradicting statements.

Reviewer #3 (Remarks to the Author):

SIGNIFICANCE:

Nonsense-mediated mRNA decay (NMD) limits the abundance of CFTR mRNA carrying nonsense alleles such as the W1282X allele. Thus, NMD represents a major limitation for other potential therapies for

cystic fibrosis (CF) caused by the W1282X mutation in the CFTR gene. Because of its location near the normal termination codon, the CFTR-W1282X truncated protein retains partial function. This has led to the suggestion that increasing its levels by inhibiting NMD will likely be beneficial. However, NMD regulates the normal expression of many genes, so an approach that utilizes gene-specific stabilization of CFTR-W1282X mRNA expression is an interesting new idea.

In this study, the authors developed a cocktail of three ASOs that restores a modest level of CFTR W1282X mRNA abundance in ASO-transfected human bronchial epithelial cells, leading to an increase in CFTR-mediated chloride current. These results suggest that allele-specific therapy for CF patients that carry the W1282X mutation may be possible. I think this is an important finding.

CRITIQUE:

This is a nice study that carefully develops a set of ASOs that inhibit NMD of CFTR mRNA carrying the W1282X allele. The results are consistent with the requirement that the PTC should occur >55 nucleotides upstream of the EJC. While this approach will likely not be possible for nonsense alleles with a large number of distal splice junctions, this is an important proof-of-concept for this gene-specific approach for NMD inhibition. I have only a few comments as indicated below.

- 1) Previous studies have shown that readthrough induced by G418 can cause a modest 1.5 to 2-fold stabilization of various nonsense-containing mRNAs. However, this seems to have been overlooked by the authors in Figure 3. I think the authors need to quantitate mRNA levels under the conditions used for these experiments (Figure 3C-L) to confirm that increasing mRNA abundance (and CFTR activity) is due solely to the ASOs, and not by G418.
- 2) In general, the activities in Figure 3 obtained are low. I think a few more controls are needed for these Ussing chamber experiments, such as: untreated (the true baseline) and G418 alone. Currently, every condition is done in the presence of the ASOs. Also, you should consider adding CFTR correctors to get as much CFTR protein out of the ER as possible and enhance the CFTR activity observed.
- 3) Page 2, Line 9: You state that restoration of as little as 10% of normal CFTR function provides a therapeutic benefit to CF patients. While true, most estimates suggest that 10%-30% restoration of CFTR function may be needed. I think that is a more reasonable range to present the range rather than the best case scenario.
- 4) The 16HBEge lines carrying the gene edited CFTR alleles express only one nonsense allele of the indicated nonsense mutations, because a selectable marker was inadvertently inserted into one CFTR allele when the cell line was originally derived by Dieter Gruenert's lab. Since you state on page 14 that the cell lines are homozygous for the indicated CFTR alleles, it implies that there are two copies of each allele. This flaw in these cell lines should be described properly.
- 5) Your abbreviations SC, LC and CC were difficult to follow initially, as they were not defined in the main text. I found them in the figure legends, but I suggest you define them at first use in the text to help the reader out.

Reviewer #1 (Remarks to the Author):

Reviewer Comment: In general, the work was done to a very high standard and the systematic exploration of sequence, length, combinations and outcomes was impressive. I have only minor suggestions for improvements.

Response: Thank you for the positive assessment and helpful suggestions.

Reviewer Comment: 1. The **figure legends** should **state the exons being targeted by the PCR primers**. This is important when comparing the outcomes of splicing analysis (Figure 2I, and Supp. Figs 7-10) with the measurements of mRNA made in Figure 2 (A-G) and Supp. Figs 2-6.

Response: We have now placed red bars above the cartoons of exons to indicate the placement of the RT-PCR primers, and made this clear in the revised legend.

Reviewer Comment: 2. There is an unfortunate disjunction in the oligonucleotides used. At the foot of page 5 it is asserted that the lead cocktail contained C495 (with C478 and C515). However, the following line refers to Supp. Fig 2 where, in panels D and G, C496 was used. Later the lead cocktail became C478+C494+C514 (Figure 2D, E and F), but Figs 2G and H, using the lead ASOs, turned out to have used C495+C515 (also in Supp. Figs 5, 6 and 8). Supp. Figs 9 and 11 used C494+C515. In all these cases the mixture is described as the lead cocktail. Figure 3 used C494+C515 (A) and C494+C514 (D). It is unlikely that the changes mattered, in terms of outcomes, but it would be helpful if the authors could find a way to either state that or indicate in the main text which 'lead cocktail' is being used.

Response: To avoid confusion, we changed the cocktail label in the revised figures, legends, and text. The cocktails are now labeled LC15-1 (C478/495/515), LC15-2 (C478/494/514), and LC15-3 (C478/494/515) and this nomenclature is used consistently in all figures and main text.

Reviewer Comment: 3. Supplementary Figure 3 has 17 bars and **18 labels**.

Response: C507 was accidentally added in the label, and we have now corrected the figure.

Reviewer Comment: 4. Panels C and D in Supp. Fig. 11 are **mislabeled**.

Response: We corrected the labels to match the panels.

Reviewer Comment: 5. The ability of **G418 to increase expression of truncated but not full-length CFTR needs some discussion.** Apart from an undetectable increase in full-length protein, is there any other reasonable explanation of its functional effects?

Response: Shortly following the pioneer round of translation, most NMD-sensitive mRNA with EJC in the 3'UTR are rapidly degraded by NMD (1, 2). The read-through activity of G418 during the pioneer round of translation could stabilize some of the *CFTR-W1282X* mRNA by removing the EJC in the 3'UTR, such that the mRNA can now undergo multiple rounds of translation to produce (mostly) truncated CFTR protein that is partially functional. The polysome-associated *CFTR-W1282X* mRNA bound to eIF4E likely would remain stable, although some proportion of it may still be targeted by NMD (1). Consistent with this model, we previously observed that G418 can inhibit NMD and promote translation of full-length protein, using NMD-sensitive *HBB-T39* mRNA (3). As previous reports (4, 5) and our data (revised Supplement Figs.14a-b) show that the read-through activity of G418 is not detectable by Western blotting, full-length CFTR protein likely contributes little to the increase in the CFTR activity.

For the revision, we measured the *CFTR* mRNA levels in the presence of G418 more carefully, by performing RT-qPCR using mRNA from the 16HBE-W1282X cells treated with the lead combination ASOs, G418, or both (revised Supplementary Fig. 13b). G418 treatment increases *CFTR* and endogenous NMD-targeted *SRSF2* mRNA levels in a dose-dependent manner (revised Supplementary Fig. 13b). The lead ASO cocktail increases *CFTR* mRNA levels without increasing *SRSF2* mRNA levels (revised Supplementary Fig. 13b). We also show that combining G418 with ASO led to a greater *CFTR* mRNA level increase than either treatment alone. This observation suggests that G418 improves CFTR activity by inhibiting NMD of *CFTR-W1282X* mRNA. However, we cannot rule out the possibility that a low level of readthrough activity, undetectable on the Western blot, contributes to the enhancement of CFTR activity. We added this discussion in the main text.

Reviewer #2

Reviewer Comment: While the study has many limitations (listed below), it is nonetheless a high-quality study demonstrating the potential of NMD-inhibition ASO approach for CF treatment. With earnest

revision efforts to mitigate some of the addressable limitations, this manuscript is considered to have enough merits to be published in Nature Communications.

Response: Thank you for the constructive criticism.

Reviewer Comment: Novelty and innovativeness: As mentioned in the manuscript (line 81), the authors had already published the concept of ASO-mediated NMD inhibition. Hence, the novelty and innovativeness of the current study are limited to the application of the concept to CF.

Response: We indeed published the GAIN strategy in 2016, so the idea of targeting the EJC with ASO to inhibit NMD is not new. However, our first paper only showed the results using minigene and chimeric model constructs, and this is the first time that we are showing the effects of GAIN ASOs in a relevant disease model, with the resulting functional improvements. Moreover, we show for the first time that GAIN is applicable to a situation that requires a cocktail of three ASOs. It is also the first time that we apply the method to CF and to a relatively frequent mutant allele. As many as 20% of hereditary diseases are caused by nonsense mutations (6), and some of them may benefit from our GAIN technology, but whether this is actually the case has to be proven on a case-by-case basis.

Reviewer Comment: Clinical applicability - delivery issues: Cystic fibrosis is a disorder that mainly affects the lung. Therefore, deliverability to the lung, precisely to bronchial epithelial cells, is one of the most critical factors for successful CF drug development. Although multiple preclinical studies have demonstrated that ASOs can be delivered to the lung as aerosols, it has not been proven clinically (no FDA-approved ASO drugs targeting the lung). Hence, the lung delivery issue remains a significant hurdle for the clinical application of this study.

Response: We agree that lung delivery of ASO is an important issue in drug development for CF, and we included a discussion of the aerosolized ASO delivery literature in the revised main text. Though aerosolized ASOs have not been approved yet, this delivery method to the lung is being actively pursued in clinical trials.

A clinically relevant method of intratracheal delivery is aerosolized ASO generated with a nebulizer (7). In mice, the bioavailability of an aerosolized MOE ASO compared to IV ASO administration was 1260% (8). Also, the reported half-lives of ASOs in the lungs of mice and

monkeys are 4 days and >7 days, respectively (8). Aerosolized gapmer ASO targeting *Scnn1a* mRNA administered to *Nedd4L* KO mice, a model of CF-like lung disease, was well tolerated up to 2 mg/kg without significant systemic exposure, and 0.1-0.3 mg/kg of the same aerosolized ASO resulted in 10 µg/g lung tissue concentration, with 70% reduction in *Scnn1a* expression (9). Currently, three inhaled ASO strategies are in active clinical trials, demonstrating the potential of aerosolized ASO as a potentially viable delivery method to target the CF airway with our lead ASO cocktail. IONIS-ENaCRx, a cET PS gapmer ASO targeting *SCNNA1*, showed in vivo activity without severe side effects in a phase-1/2a clinical trial that recently completed (8). An ASO cocktail developed by Topigen, TPI ASM8 simultaneously targets CCR-3, IL-3, IL-5, and GMCSF to treat asthma using two PS ASOs, TOP004 and TOP005 (10). Its phase-2 clinical trial was completed in 2013 and showed target gene knockdown in vivo, without severe side effects (8). The TPI ASM8 results suggest that ASO cocktails are potentially a viable therapeutic strategy. QR-010, a uniformly 2'-OMe PS modified ASO developed by ProQR and currently in a phase-2 clinical trial, inserts the three missing bases in the mutant *CFTR-F508del* mRNA, converting it into WT *CFTR* mRNA through an unknown mechanism in vivo (11). The results of these studies will provide useful information about nucleic-acid therapeutics for CF and other lung diseases.

Reviewer Comment: Clinical applicability - issues related to cocktail treatment: The authors showed that the lead ASO cocktail is effective when all three ASOs are administered at the same time and with a specific stoichiometry (1:1:1 ratio). This requirement poses two major challenges for the clinical application. The first challenge pertains to efficacy. It is not given that the three ASOs have equal in vivo distribution and cellular uptake rates. The potential difference in the distribution/uptake rates may lead to skewed stoichiometry deviating from the optimal ratio and, hence, diminished efficacy. The second challenge pertains to tolerability. A cocktail of three ASOs comes with three times the risk of unexpected toxicity as monotherapy with a single ASO. Even if the authors demonstrate that the three ASOs do not have critical off-targets (by sequence alignments or RNA-seq), it does not preclude that at least one of three ASOs have some unpredictable toxicity. The higher risk of toxicity may hamper the clinical application of the study.

Response: Optimization of the stoichiometry of the ASO cocktail and potential toxicity of the ASO cocktail are important factors to consider for lead compound optimization. We agree that individual ASOs in the cocktail may have different distribution and uptake, thus the need for optimization of the stoichiometry, as we have done, and this should probably be revisited during

further preclinical and clinical development. However, cocktail drugs are not necessarily toxic if they are carefully studied, as demonstrated by the efficacy and safety of Trikafta, which is composed of three CFTR modulators, VX-770/661/445.

While we used a 1:1:1 ratio of the ASO cocktail in many experiments, we showed other ASO cocktails with lower total ASO concentration and different ratios (e.g., 1:2:2 and 4:4:1) that inhibited NMD at statistically non-distinguishable levels compared to the 1:1:1 ASO cocktail (Fig. 2h). We also identified in our ASO screens various other combinations that gave similar degrees of increase in *CFTR* mRNA levels in 16HBE-W1282X cells (Fig. 2a-c). Thus, there is considerable flexibility if toxicity or tolerability issues were to arise.

Given that the chance of binding to an off-target for an ASO with ≥ 4 -nt mismatches is very low (12), we believe the chance of hybridization-mediated off-target effect from the ASO cocktail is low. However, we agree with the reviewer that sequence alignment or RNA-seq experiments do not preclude unexpected off-target effects. Although RNA-seq can identify off-target effect candidates, we have not relied on it, in part because this would not detect ASOs that might inhibit translation, for example; conversely, an off-target effect on splicing or RNA stability is not necessarily deleterious.

We believe that systematic toxicology studies in rodents and non-human primates—a required step in clinical development—are a more effective way to nominate a clinical candidate. We previously followed this approach for Spinraza, and are currently following it for an ASO for familial dysautonomia. Nevertheless, we previously studied hybridization-mediated off-target effects of splice-switching ASOs and how to mitigate them, looking at the effects of ASO length, chemistry, delivery method, etc. (13). Relevant to the present study, using 18mer ASOs delivered by free uptake greatly reduced off-target effects.

Reviewer Comment: Functional evidence: It's not clear if the extent of CFTR functional restoration shown in Fig 3 is enough to produce symptomatic benefit in patients. The authors did not mention how much increase in the CFTR-mediated chloride current should be enough for symptomatic benefit. The authors did mention that "As low as 10% of normal CFTR function provides a significant therapeutic benefit for CF patients", but this "10%" refers to the gene expression level, not the chloride current level. Also, the authors did not mention whether or not the observed increase in the chloride current by the ASO cocktail corresponds to more than 10% of the normal current level (the normal current level, an essential reference, is not provided in this study). Judging from the fact that the ASO cocktail led to only

a low level of increase of the partially functional (with skipped exons and readthrough) protein (Fig. 3A), the level of functional restoration could be lower than 10%.

Response: We agree that the 10% in “as low as 10% of normal CFTR function provides a significant therapeutic benefit for CF patients” technically refers to the gene expression level. Also, as Reviewer 3 pointed out, the CFTR activity threshold above which clinical benefit is conferred is a range rather than a single number, and we made it clear in the revised main text that 10%-30% CFTR activity compared to WT CFTR is beneficial.

To understand the relative activity of CFTR-W1282X protein compared to wild-type CFTR protein (CFTR-WT), we generated 16HBE-W1282X cells with doxycycline (dox)-inducible overexpression of wild-type CFTR-WT or CFTR-W1282X (dubbed 16HBEge-GFP-P2A-WT or 16HBEge-GFP-P2A-W1282X), using a lentiviral vector that allows co-expression of Turbo GFP via a 2A self-cleaving peptide (revised Fig. 4). The induction of the recombinant gene by dox was confirmed by fluorescence microscopy (revised Fig. 4b). Since the Turbo GFP and CFTR proteins are translated initially as the same polypeptide chain, GFP levels can be used to control expression from the integrated plasmid (14). Treating the 16HBEge-GFP-P2A-WT/W1282X cells with 2 μ g/mL dox resulted in similar levels of GFP signal, indicating that the expression of recombinant CFTR proteins is induced at similar levels (revised Fig. 4c). We verified the expression of recombinant CFTR-WT and CFTR-W1282X in 16HBEge-GFP-P2A-WT/W1282X cells treated with or without dox (revised Supplementary Fig. 12).

16HBEge-GFP-P2A-WT or 16HBEge-GFP-P2A-W1282X cells treated with dox and Trikafta (VX-445, VX-661, and VX-770) showed significant increases in chloride current compared to Trikafta-only treatment, as quantified by the relative total area under the curve normalized to the total AUC after acute VX-770 treatment, but before the CFTR inhibitor 172 treatment (revised Fig. 4d-g). The total AUC of 16HBEge-GFP-P2A-W1282X after VX-770 treatment was about 42% and 28% of the total AUC before and after VX-770 treatment in 16HBEge-GFP-P2A-WT cells, respectively (revised Fig. 4d and 4e). The background CFTR activity in 16HBE-W1282X cells due to endogenous CFTR-W1282X expression is negligible, as the chloride currents of the no-dox-treatment 16HBEge-GFP-P2A-WT/W1282X cells were substantially lower than those of the dox-treated cells (revised Fig. 4f and 4g).

We tested the effect of lead ASO cocktail LC15-2 co-treated with Trikafta. Compared to the scramble ASO and Trikafta co-treatment and all other treatments we previously reported, LC15-

2 and Trikafta co-treatment led to a significant CFTR activity enhancement. Also, we compared the Isc response to the acute treatment with inhibitor 172 of our experiment to that in a previously published paper that characterized the CFTR activity in parental 16HBE14o- cell (4). From this, we estimate that the combination treatment with the lead ASO cocktail and Trikafta results in relative CFTR-W1282X activity that is approximately 18% to 30% compared to WT-CFTR, consistent with the results from 16HBEge-GFP-P2A-WT/W1282X cells. Thus, we conclude that CFTR-W1282X is partially functional, and the level of CFTR activity enhancement we achieved with the lead ASO cocktail and Trikafta co-treatment may be within the desirable range.

Reviewer Comment: Potential side effect: The authors showed that the ASO cocktail leads to undesirable exon skipping events. These exon skipping events can adversely impact the function of CFTR protein, especially when a patient treated with the cocktail is compound heterozygous for the W1282X mutation. Since the ASO cocktail can further damage the function of the other allele by inducing exon skipping, the allele might not be able to function properly even with the help of other CF drugs, such as VX770, VX809, and/or G418. Thus, it would be helpful to demonstrate that the ASO cocktail does not diminish the CFTR function in bronchial epithelial cells of normal individuals and CF patients with non-W1282X mutations.

Response: We agree that the unexpected splicing changes involving exons 24-26 mean that our ASOs should not be used for CFTR mutations other than CFTR-W1282X. The effect of the splicing changes in these exons is not clear, as we have not tested the ASOs on the parental 16HBE14o-cells with WT CFTR. We believe that our GAIN technology is best suited for homozygous W1282X/W1282X mutations or compound heterozygous mutations with W1282X and another severe allele not amenable to currently approved therapies. We have revised the Discussion section accordingly.

Reviewer Comment: ESE prediction: The authors used ESEfinder to identify ESEs, only to dismiss the ESE prediction result as inconclusive because it is only a prediction (Line 203-205). Perplexingly, the authors then reference the prediction results to interpret some of the exon skipping events. The authors may consider clarifying these seemingly self-contradicting statements.

Response: *CFTR* exons 24, 25, and 26 are predicted to harbor several ESEs. We tried to **predict** putative ESEs by using ESEfinder, which searches for SR protein motifs along a sequence. The reviewer's comment helped us realize that we should have stated that we used ESEfinder to "predict putative ESEs on *CFTR* exons 24-26" rather than to "identify putative ESEs on *CFTR* exons 24-26". Identification of an actual ESE that regulates the splicing of an exon requires experimental validation, but we used ESEfinder only to look for putative ESEs. We also added an explanation to the statement, "overlap with an SR protein motif is insufficient to predict an ASO's interference with splicing" (initial submission line 204-205). Some of the putative ESEs on exon 24 and 25 predicted by ESEfinder overlap with the target sites of ASOs C478 and C494/495, which induced exon 24 and 25 skipping, respectively. This overlap suggests that these ASOs may induce exon skipping by preventing the binding of splicing factors at these putative ESEs. However, ESEfinder may report false positives. For example, the putative SRSF1 (SF2/ASF), SRSF2 (SC35), and SRSF5 (SRp40) binding sites on exon 26 predicted by the ESEfinder overlap extensively with some of the 15-mer exon 26-targeting ASOs (C508-C526) (revised Supplementary Fig. 7b). Among them, C514 and C515 are used in the lead ASO cocktails LC15-2 and LC15-1, respectively. None of the ASOs C508-C526 induced detectable exon 26 skipping in DLD1 cells; neither C515 nor C26, an 18-mer lead ASO that covers regions targeted by C514 and C515, individually caused exon skipping in 16HBE-W1282X cells, even though a substantial portion of the SRSF5 (SRp40) binding motif overlaps with the target site. Some ASOs among C508-C526 may fail to promote exon skipping because they block the binding of two or more splicing factors that offset each other's effects. In this scenario, some of the ASOs that only target splicing factors binding to ESEs could promote exon skipping. However, as none of the ASOs disturbed exon 26 splicing, at least some of the ESEs on exon 26 predicted by ESEfinder could be false positives.

For clarification of these points, we added to the revised main text "mRNA sequences complementary to the lead ASOs C478 and C494 that caused exon 24 and 25-skipping overlap with SR protein motifs predicted by ESEfinder; however, the target sites for the lead ASOs C514 and C515, which did not cause any exon 26 skipping, overlap with an SRSF5/SRp40 motif. Thus, overlap with an SR protein motif is insufficient to predict an ASO's interference with splicing."

We are not sure which section the reviewer refers to in the comment "Perplexingly, the authors then reference the prediction results to interpret some of the exon skipping events," as we did

not reference the prediction results after line 205. After line 205, we mention ESEs in the following context: “C478 is unlikely to cause exon 25 and 26 skipping by binding to ESEs in these exons. These results suggest that an ESE and/or the EJC in exon 24 is involved in long-range splicing regulation.” The ESEs in these sentences are not referring specifically to those predicted by ESEfinder, but rather to any actual ESE on exon 25 and 26 that regulates splicing. ESEfinder may predict some but not all of these ESEs. In the above sentences, we intended to convey that the chance of ASO C478 binding to exon 25 or 26 is low, based on the sequence alignment; if C478 is unlikely to bind to exon 25 or 26, then the exon-skipping events involving these exons and resulting from the high-dose C478 treatment are unlikely to be caused by direct disruption of splicing-factor binding to any ESE on exon 25 or 26.

Reviewer # 3

Reviewer Comment: This is a nice study that carefully develops a set of ASOs that inhibit NMD of CFTR mRNA carrying the W1282X allele. The results are consistent with the requirement that the PTC should occur >55 nucleotides upstream of the EJC. While this approach will likely not be possible for nonsense alleles with a large number of distal splice junctions, this is an important proof-of-concept for this gene-specific approach for NMD inhibition. I have only a few comments as indicated below.

Responses: We thank the reviewer for the positive assessment.

Reviewer Comment: 1) Previous studies have shown that readthrough induced by G418 can cause a modest 1.5 to 2-fold stabilization of various nonsense-containing mRNAs. However, this seems to have been overlooked by the authors in Figure 3. I think the authors need to quantitate mRNA levels under the conditions used for these experiments (Figure 3C-L) to confirm that increasing mRNA abundance (and CFTR activity) is due solely to the ASOs, and not by G418.

Response: We appreciate the suggestion. Although we did not show changes in *CFTR* mRNA levels in the initial submission, we were aware that G418 can inhibit NMD, and showed that G418 treatment alone can increase CFTR-W1282X protein, demonstrating that G418 inhibits NMD (Supplementary Figure 11D-E in the initial submission, revised Supplementary Fig. 14a-b). However, we agree with the reviewer that a more careful investigation of the *CFTR* mRNA is

warranted. As we discussed above in response to Reviewer 1's comment, we now show the levels of *CFTR-W1282X* mRNA and NMD-targeted *SRSF2* mRNA levels in 16HBE-W1282X cells treated with the ASOs, G418, or both together (revised Supplementary Fig. 13b). G418 treatment caused a significant increase in *CFTR* mRNA, compared to the no-treatment or scramble-ASO controls. The lead 18-mer ASO cocktail (LC18) + 200/600 μ M G418 treatment caused a statistically significant increase in CFTR activity, compared to the 18-mer scramble (Sc18) ASO + 200 μ M G418 treatment. Compared to the *CFTR* mRNA levels with 18-mer scramble (Sc18) ASO + 200 μ M G418 treatment, the lead 18-mer ASO cocktail (LC18) + 600 μ M G418 treatments resulted in significantly increased *CFTR* mRNA levels; conversely, the lead 18-mer ASO cocktail (LC18) + 200 μ M G418 increased the *CFTR* mRNA, but not significantly. These results show that the difference in CFTR activity between the lead 18-mer ASO cocktail (LC18) + G418 treatment and the 18-mer scramble ASO (Sc18) + G418 treatment is due to NMD inhibition by both the EJC-targeting ASO and by G418. These results are discussed in the revised main text.

Reviewer Comment: 2) In general, the activities in Figure 3 obtained are low. I think a few more controls are needed for these Ussing chamber experiments, such as: untreated (the true baseline) and G418 alone. Currently, every condition is done in the presence of the ASOs. Also, you should consider adding CFTR correctors to get as much CFTR protein out of the ER as possible and enhance the CFTR activity observed.

Response: We have now obtained the baseline CFTR activity with no ASO treatment or G418 treatment only. The baseline CFTR activity with VX-809 and VX-770 treatment was similar to that of 15/18-mer scramble ASO (Sc15/18) + VX-809/VX-770 treatment. We also conducted an experiment in which the cells were treated with the ASO and Trikafta (VX-770/661/445). Trikafta significantly augmented the effect of the lead ASO cocktail, compared to Orkambi (VX-770/809). These data are now described in the revised Results (revised Figure 5) and Discussion sections.

Reviewer Comment: 3) Page 2, Line 9: You state that restoration of as little as 10% of normal CFTR function provides a therapeutic benefit to CF patients. While true, most estimates suggest that 10%-30% restoration of CFTR function may be needed. I think that is a more reasonable range to present the range rather than the best-case scenario.

Response: Thank you for pointing this out. We made it clear in the revised main text that 10%-30% CFTR activity compared to WT CFTR is beneficial.

Reviewer Comment: 4) The 16HBEge lines carrying the gene edited CFTR alleles express only one nonsense allele of the indicated nonsense mutations, because a selectable marker was inadvertently inserted into one CFTR allele when the cell line was originally derived by Dieter Gruenert's lab. Since you state on page 14 that the cell lines are homozygous for the indicated CFTR alleles, it implies that there are two copies of each allele. This flaw in these cell lines should be described properly.

Response: Thank you for pointing this out. As published before (4), the 16HBEge cell lines contain SV40 genomic sequence, which was used to immortalize the parental 16HBE14o- cells, within intron 6 of one *CFTR* allele. The precise effect of this insertion on transcription or splicing of the transcript is unknown, but it was reported to make the cell line functionally monoallelic. We have added text to this effect in the revised main text.

Reviewer Comment: 5) Your abbreviations SC, LC and CC were difficult to follow initially, as they were not defined in the main text. I found them in the figure legends, but I suggest you define them at first use in the text to help the reader out.

Response: We clarified the labels in the main text, as well as in the figures.

References

1. T. Kurosaki, M. W. Popp, L. E. Maquat, Quality and quantity control of gene expression by nonsense-mediated mRNA decay. *Nat. Rev. Mol. Cell Biol.* **20**, 406–420 (2019).
2. T. A. Hoek, *et al.*, Single-Molecule Imaging Uncovers Rules Governing Nonsense-Mediated mRNA Decay. *Mol. Cell* **75**, 324-339.e11 (2019).
3. T. T. Nomakuchi, F. Rigo, I. Aznarez, A. R. Krainer, Antisense oligonucleotide-directed inhibition of nonsense-mediated mRNA decay. *Nat Biotech* **34**, 164–166 (2016).
4. H. C. Valley, *et al.*, Isogenic cell models of cystic fibrosis-causing variants in natively expressing pulmonary epithelial cells. *J. Cyst. Fibros.*, 8–15 (2018).

5. M. M. Keenan, *et al.*, Nonsense-mediated RNA decay pathway inhibition restores expression and function of W1282X CFTR. *Am. J. Respir. Cell Mol. Biol.* **61**, 290–300 (2019).
6. M. Mort, D. Ivanov, D. N. Cooper, N. A. Chuzhanova, A meta-analysis of nonsense mutations causing human genetic disease. *Hum. Mutat.* **29**, 1037–1047 (2008).
7. M. V. Templin, *et al.*, Pharmacokinetic and toxicity profile of a phosphorothioate oligonucleotide following inhalation delivery to lung in mice. *Antisense Nucleic Acid Drug Dev.* **10**, 359–368 (2000).
8. S. T. Crooke, *Antisense drug technology: principles, strategies, and applications* (CRC press, 2007).
9. J. R. Crosby, *et al.*, Inhaled ENaC antisense oligonucleotide ameliorates cystic fibrosis-like lung disease in mice. *J. Cyst. Fibros.* **16**, 671–680 (2017).
10. Z. Allakhverdi, M. Allam, P. M. Renzi, Inhibition of antigen-induced eosinophilia and airway hyperresponsiveness by antisense oligonucleotides directed against the common β chain of IL-3, IL-5, GM-CSF receptors in a rat model of allergic asthma. *Am. J. Respir. Crit. Care Med.* **165**, 1015–1021 (2002).
11. P. C. Zamecnik, M. K. Raychowdhury, D. R. Tabatadze, H. F. Cantiello, Reversal of cystic fibrosis phenotype in a cultured Delta508 cystic fibrosis transmembrane conductance regulator cell line by oligonucleotide insertion. *Proc. Natl. Acad. Sci. U. S. A.* **101**, 8150–8155 (2004).
12. T. Yoshida, *et al.*, Evaluation of off-target effects of gapmer antisense oligonucleotides using human cells. *Genes to Cells* **24**, 827–835 (2019).
13. J. Scharner, *et al.*, Hybridization-mediated off-target effects of splice-switching antisense oligonucleotides. *Nucleic Acids Res.*, 1–15 (2019).
14. D. Wu, P. A. Yates, H. Zhang, K. Cao, Comparing lamin proteins post-translational relative stability using a 2A peptide-based system reveals elevated resistance of progerin to cellular degradation. *Nucleus* **7**, 585–596 (2016).

REVIEWER COMMENTS

Reviewer #1 (Remarks to the Author):

The revisions made by the authors satisfy my minor suggestions.

Reviewer #2 (Remarks to the Author):

Remaining issues:

(1) To argue that the ASO therapy can enhance CFTR activity up to a therapeutic range, the authors compared the activity of the wild type protein vs. the W1282X protein. Based on the observation that the W1282X protein has 28-42% of the activity of the wild type protein, the authors concluded that "This result suggests that the combination treatment may enhance CFTR activity up to a therapeutic range." However, this conclusion holds only when the ASO therapy inhibits the NMD perfectly and does not induce undesirable exon skipping; due to the incompleteness of ASO-mediated NMD inhibition and undesirable exon skipping effects, the protein level from the W1282X allele could be much lower than the protein level from the wild type allele (as seen in Fig. 3A). In addition, given that at least 10% of the wild type CFTR activity is required for phenotypic rescue and that the ASO therapy acts on only one allele, if the therapy is used for patients with only one W1282X allele (compound het), the therapy needs to be able to enhance the CFTR activity up to at least 20% of the wild type CFTR activity from that W1282X allele to achieve phenotypic rescue. The authors need to provide clarifications for these points in order not to overstate the therapeutic potential of the ASO therapy.

(2) The authors noted that "We estimate that the combination treatment with the lead ASO cocktail and Trikafta results in the CFTR-W1282X activity that is approximately 18% to 30% relative to the previously reported wild-type CFTR activity in 16HBE14o- cells (17)". However, in making this estimation (a critical estimation to justify clinical significance of this study), the authors did not measure the wild type cell CFTR activity, and instead borrowed the measurement from some other study. Comparing activity levels measured by two different studies unlikely yields robust estimation, and therefore the 18-30% estimate is not credible. In order to firmly establish the estimation of CFTR functional restoration by the ASO therapy, the CFTR activity in wild type cells should be measured by the authors themselves with the same methods and conditions together when measuring the CFTR activity in CFTR-mutant cells with the ASO treatment.

(3) The examples and references that the authors used to justify that ASO therapy for lung diseases is readily applicable to clinic seem incorrect.

1) The clinical trial for IONIS-ENaC-2.5-Rx is inactive because they found evidence of toxicity from a long-term follow up of a preclinical toxicology study. The authors used an irrelevant reference that even pre-dates the IONIS-ENaC-2.5-Rx trial (from the year 2007).

2) TPI ASM8 did not show statistical significance for the primary endpoint. An irrelevant reference was used here too (the same review paper as above, from the year 2007).

3) For QR-010, the authors need to provide evidence that it is currently in a phase 2 clinical trial, as I could not find it from clinicaltrials.gov.

(4) The authors kept mentioning that the ASO therapy is "allele-specific" in the abstract and the main text. However, it seems mis-leading because the ASO therapy actually can act non-allele-specifically. It acts on both alleles, but it helps a certain specific mutant alleles, and it hurts the other alleles (by inducing undesirable exon skipping). As the authors pointed out, the ASO therapy is recommended only for patients who are homozygous or essentially hemizygous (e.g., the other allele is deleted) for certain mutations, such as W1282X. Therefore, it feels mis-leading to call this therapy as "allele-specific."

Reviewer #3 (Remarks to the Author):

Low CFTR mRNA expression due to nonsense-mediated mRNA decay (NMD) is a major hurdle in developing a therapy for cystic fibrosis (CF) caused by the W1282X mutation in the CFTR gene. CFTR-W1282X truncated protein retains partial function, so increasing its levels by inhibiting NMD of its mRNA will likely be beneficial. Because NMD regulates the normal expression of many genes, gene-specific stabilization of CFTR-W1282X mRNA expression is more desirable than general NMD inhibition. Synthetic antisense oligonucleotides (ASOs) designed to prevent binding of exon junction complexes (EJC) downstream of premature termination codons (PTCs) attenuate NMD in a gene-specific manner. The authors developed a cocktail of three ASOs that specifically increases the expression of CFTR W1282X mRNA and CFTR protein in ASO-transfected human bronchial epithelial cells. This treatment increased the CFTR-mediated chloride current. Overall, this study sets the stage for clinical development of an allele-specific therapy for CF caused by the W1282X mutation.

The authors have addressed all of my prior concerns. Congratulations on this nice study!

Reviewer #1 (Remarks to the Author):

The revisions made by the authors satisfy my minor suggestions.

Authors response> Thank you for the helpful feedback.

Reviewer #2 (Remarks to the Author):

Remaining issues:

(1) To argue that the ASO therapy can enhance CFTR activity up to a therapeutic range, the authors compared the activity of the wild type protein vs. the W1282X protein. Based on the observation that the W1282X protein has 28-42% of the activity of the wild type protein, the authors concluded that "This result suggests that the combination treatment may enhance CFTR activity up to a therapeutic range." However, this conclusion holds only when the ASO therapy inhibits the NMD perfectly and does not induce undesirable exon skipping; due to the incompleteness of ASO-mediated NMD inhibition and undesirable exon skipping effects, the protein level from the W1282X allele could be much lower than the protein level from the wild type allele (as seen in Fig. 3A).

In addition, given that at least 10% of the wild type CFTR activity is required for phenotypic rescue and that the ASO therapy acts on only one allele, if the therapy is used for patients with only one W1282X allele (compound het), the therapy needs to be able to enhance the CFTR activity up to at least 20% of the wild type CFTR activity from that W1282X allele to achieve phenotypic rescue. The authors need to provide clarifications for these points in order not to overstate the therapeutic potential of the ASO therapy.

Authors response> We agree with the reviewer that the Discussion should be written to minimize any overstatement of the therapeutic potential of the ASO therapy.

Please note that the quoted statement appears in the following context in the Discussion section: "The combined treatment of the GAIN ASO cocktail and Trikafta in human bronchial epithelial cells with the W1282X mutation enhanced CFTR activity to 10-30% of wild-type CFTR. This result *suggests* that the combination treatment *may* enhance CFTR activity up to a therapeutic range. Consistent with this *possibility*, we estimate that the intrinsic activity of CFTR-W1282X protein, controlling for the plasmid expression levels, is approximately 28 – 42% of CFTR-WT protein, when combined with Trikafta."

We believe that the three words shown above in italics (*suggests*, *may*, *possibility*) clearly indicate that we are being appropriately cautious, rather than stating a definitive conclusion. In addition, the statement, "This result suggests that the combination treatment may enhance CFTR activity up to a therapeutic range," appears after the sentence "The combined treatment of the GAIN ASO cocktail and Trikafta in human bronchial epithelial cells with the W1282X mutation enhanced CFTR activity to 10-30% of wild-type CFTR." Thus, we did not conclude that GAIN technology could be therapeutically beneficial based on the 28-42% relative intrinsic activity of the protein, which is mentioned in the sentence after that. We mentioned the 28-42% figure to illustrate that increasing the CFTR-W1282X protein expression in the presence of Trikafta can lead to a significant increase in CFTR activity.

We agree with the reviewer that the relative activity of 28-42% could be achieved with the ASO cocktail treatment only if NMD in 16HBE-W1282X cells is inhibited by nearly 100%. However, we did not claim that the ASO cocktail increases the CFTR-W1282X activity to 28-42% of CFTR-WT activity. As mentioned

above, we estimated indirectly that the ASO cocktail combined with Trikafta could increase CFTR-W1282X activity in 16HBE-W1282X cells to 18-30% of CFTR-WT in 16HBE14o- cells.

Please note also that the exon-skipping activity of the ASO cocktail would not change the amino-acid sequence of CFTR-W1282X, because we showed that the exon-skipping events occur in the 3' UTR (i.e., exons 24-26) of the *CFTR*-W1282X mRNA. Alternative 3'UTRs can potentially affect the translation efficiency of the *CFTR*-W1282X mRNA, but the impact of exon-skipping events in exon 24-26 on translation cannot be predicted from the sequence information only. We have now made this point clear by including the following statement in the Results section: "The exon-skipping events occurred in the 3'UTR of the *CFTR*-W1282X mRNA, and thus should not affect the amino-acid sequence of CFTR-W1282X protein. The length and sequence variation in the resulting 3'UTRs could have a positive, negative, or no effect on translation efficiency."

As we had mentioned in the Discussion section, "our GAIN technology might be best suited for homozygous W1282X/W1282X mutations or compound heterozygous mutations with W1282X and a severe allele not amenable to currently approved therapies," due to the the exon-skipping activity. However, we performed these experiments in the 16HBE-W1282X cells, which only express the W1282X allele, and the ASO cocktail improved CFTR function in these cells, in spite of the exon-skipping activity. Note that the 16HBE14o- parental cells and 16HBE-W1282X cells are functionally monoallelic due to the SV40 sequence in one of the *CFTR* alleles, as we mentioned in the Results section ¹.

The reviewer also pointed out that the protein expression level in the CFTR-W1282X cells is lower than the protein level from the wild-type allele in Fig. 3A. In this figure, the CFTR-WT protein was from the wild-type DLD1 colorectal cancer cell line, which has a different lineage from the 16HBE-W1282X cell line. However, we do not believe that this is a cell-line-specific effect, because doxycycline-induced CFTR-W1282X expression in 16HBE-P2A-W1282X cells is significantly less than doxycycline-induced CFTR-WT expression in 16HBE-P2A-WT cells (Supplemental Fig. 12). This is attributable to the known instability of the CFTR-W1282X protein, and we have now clarified this point in the Results section with the following statement: "The reduced level of CFTR-W1282X compared to CFTR-WT is attributable to the lack of C-terminal residues (¹⁴⁷⁸TRL¹⁴⁸⁰) important for post-translational processing and stability of CFTR protein ^{2, 3}". We demonstrated that 16HBE-P2A-W1282X cells treated with doxycycline and Trikafta could achieve 28-42% relative CFTR activity, compared to 16HBE-P2A-WT cells expressing CFTR-WT protein, despite the lower CFTR-W1282X protein levels. Thus, we do not believe that similar CFTR protein levels between WT and W1282X are required to achieve a significant enhancement of CFTR activity.

We agree with the reviewer that our therapeutic strategy needs to enhance CFTR activity from the W1282X allele up to at least 20% of wild-type CFTR activity, to achieve phenotypic rescue in a compound heterozygous cell. We have accordingly modified the relevant passage in the Discussion section (new text shown here in italics): "Thus, our GAIN technology might be best suited for homozygous W1282X/W1282X mutations or compound heterozygous mutations with W1282X and a severe allele not amenable to currently approved therapies. *For the GAIN technology to be applied to these compound heterozygous mutations, CFTR activity from the W1282X allele would have to increase to at least 20% of the wild-type CFTR activity to achieve phenotypic rescue. In the future, testing the GAIN ASO cocktail in a humanized mouse model with the W1282X mutation and patient-derived airway epithelial models could further help to evaluate its therapeutic potential.*"

(2) The authors noted that "We estimate that the combination treatment with the lead ASO cocktail and Trikafta results in the CFTR-W1282X activity that is approximately 18% to 30% relative to the previously reported wild-type CFTR activity in 16HBE14o- cells (17)". However, in making this estimation (a critical estimation to justify the clinical significance of this study), the authors did not measure the wild-type cell CFTR activity, and instead borrowed the measurement from some other study. Comparing activity levels measured by two different studies unlikely yields robust estimation, and therefore the 18-30% estimate is not credible. In order to firmly establish the estimation of CFTR functional restoration by the ASO therapy, the CFTR activity in wild type cells should be measured by the authors themselves with the same methods and conditions together when measuring the CFTR activity in CFTR-mutant cells with the ASO treatment.

Authors response>

Firstly, we agree that the assessment of the therapeutic potential of the ASO cocktail should be conducted carefully. However, we are not arguing that our data are sufficient to justify immediate initiation of a clinical trial. Our CFTR activity estimation using the 16HBE cells is only an estimate, but it provides encouraging support in this proof-of-concept study. As we mentioned in the Discussion section, more clinically relevant information should be acquired from patient-derived cells and in vivo models (e.g., humanized mouse models). We would not expect the relative rescue of CFTR activity to be identical in all these settings, so even the numbers obtained in a side-by-side comparison of wild-type and mutant 16HBE cells should not be overinterpreted.

The commercial parental cells are very expensive (\$1,440/vial), and the license for use and material transfer agreement are burdensome, which would significantly delay the publication of our study. Instead, to make our indirect comparison between the 16HBE-W1282X cells and the parental 16HBE14o-cells as valid as possible, we consulted very closely with the authors of the original study, so that we could conduct the Ussing chamber experiments in the same way they did. The parameters we discussed with them included: cell-culture conditions, buffers used during the Ussing Chamber assay, instrumentation, and CFTR-targeting drugs added during the experiments. An automated robotic instrument was used for the short-circuit current measurement in the original study; however, the authors confirmed to us that the CFTR activities measured by automated and non-automated methods are equivalent. Thus, the key aspects of the experiments were equivalent between the original study and ours. Note that we acknowledged the authors of that study generously sharing protocols and advice, and for helpful comments on the manuscript.

(3) The examples and references that the authors used to justify that ASO therapy for lung diseases is readily applicable to clinic seem incorrect.

1) The clinical trial for IONIS-ENaC-2.5-Rx is inactive because they found evidence of toxicity from a long-term follow up of a preclinical toxicology study. The authors used an irrelevant reference that even pre-dates the IONIS-ENaC-2.5-Rx trial (from the year 2007).

2) TPI ASM8 did not show statistical significance for the primary endpoint. An irrelevant reference was used here too (the same review paper as above, from the year 2007).

3) For QR-010, the authors need to provide evidence that it is currently in a phase 2 clinical trial, as I could not find it from clinicaltrials.gov.

Author response> We thank the reviewer for the feedback about the references. We have updated the references for IONIS-ENaC-2.5-Rx and TPI ASM8 ^{4,5}. We originally used the Crooke 2007 reference for TPI ASM8, as it reviews the mechanism of TPI ASM8, but we agree there are more appropriate references. Before submission we had a different reference for IONIS-ENaC-2.5-Rx, but to avoid

exceeding 70 references, we tended to favor review articles, and inappropriately used Crooke 2007 in this case. QR-010 completed phase-1b of a phase-1/2 clinical trial (NCT02532764), and we now corrected this in the Discussion section.

We agree with the reviewer that the aerosolized ASO examples mentioned above have limitations and have not yet been approved by the FDA. Developing aerosolized ASOs is not trivial, and delivery, efficacy, and toxicity should be assessed and optimized through in vivo experiments, as we had mentioned in the Discussion. Despite their limitations, these studies demonstrated that naked aerosolized ASOs can be delivered to the airway tissue. IONIS-ENaC-2.5-Rx and TPI ASM8 knocked down their intended target genes, and in a phase-1b clinical trial (NCT02564354), QR-010 was well tolerated and improved chloride transport in human subjects ⁴⁻⁶.

We made the following revisions to indicate the limitations of these studies.

1) We revised the original statement "Several aerosolized ASOs tested in clinical trials provide a useful precedent for a viable delivery method for targeting the CF airway with our lead ASO cocktail" to: "Several clinical trials have already shown that aerosolized ASOs can be efficiently delivered to the airway tissues, although further studies to demonstrate safety and efficacy will be essential."

2) We revised the original sentence "Ionis Pharmaceuticals developed an aerosolized gapmer ASO targeting *SCNNA1*, which had significant in vivo activity and no severe side effects in a phase-1/2a clinical trial" to the following: " Ionis Pharmaceuticals developed an aerosolized gapmer ASO, IONIS-ENaC-2.5-Rx, targeting *SCNNA1*. It showed significant in vivo activity and no safety concerns in a phase-1/2a clinical trial (NCT03647228); however, due to long-term toxicity observed in the preclinical model, its development was recently discontinued ⁴."

3) We also revised the original sentence "TPI ASM8 knocked down its target genes with no severe side effects in a phase-2 clinical trial" to: "TPI ASM8 knocked down its target genes efficiently in rodents and non-human primates ⁵. Phase-2 clinical trials (NCT01158898, NCT00264966, and NCT00822861) demonstrated the safety of TPI ASM8 in humans. The reduction in late-phase allergen response (primary outcome measure) between TPI ASM8 and placebo was statistically significant in the open-label study (NCT00822861), but not in a double-blinded placebo-controlled study (NCT01158898) ⁷; however, TPI ASM8 reduced inflammatory response and sputum eosinophils in dose-escalation studies ⁵."

(4) The authors kept mentioning that the ASO therapy is "allele-specific" in the abstract and the main text. However, it seems mis-leading because the ASO therapy actually can act non-allele-specifically. It acts on both alleles, but it helps a certain specific mutant alleles, and it hurts the other alleles (by inducing undesirable exon skipping). As the authors pointed out, the ASO therapy is recommended only for patients who are homozygous or essentially hemizygous (e.g., the other allele is deleted) for certain mutations, such as W1282X. Therefore, it feels mis-leading to call this therapy as "allele-specific."

Author response> It is true that our ASO cocktail can act on other *CFTR* alleles in the sense that it can still promote exon skipping in exons 24-26. However, it is allele-specific in the sense that it is realistically beneficial only for improving *CFTR* function in the context of the *CFTR*-W1282X mutation. The ASO cocktail works by preventing exon-junction complex binding on exons 24-26. Our data presented on Fig. 2 demonstrates that the ASO cocktail stabilized the W1282X mRNA and improved function only when the ASOs targeted all three EJC binding sites, and it did not stabilize *CFTR*-G542X or *CFTR*-R1162X mRNAs

with nonsense mutations in exons 12 and 22, respectively. The ASO cocktail is expected to inhibit NMD of *CFTR* mRNAs with nonsense mutations in exons 24 (e.g., Q1313X) or 25 (e.g., Q1330X and E1371X), but these mutations are very rare (allele frequency < 0.05%) and are expected to require fewer ASOs than the three-ASO cocktail designed for the W1282X mutation: based on the predicted ribosomal footprint, Q1313X and Q1330X would need a two-ASO cocktail, and E1371X would need only one ASO. The ASO cocktail would not be effective for any missense mutation. Thus, the ASO cocktail is clearly not applicable to a broad range of *CFTR* mutations; it applies to a particular allele, making it allele-specific.

To avoid any confusion, we revised the original paragraph (lines 330-341) “The lead GAIN ASO cocktails inhibited NMD... inhibition of other mRNA-degradation pathways” to the following: “Consistent with the EJC-centric model of NMD, our results demonstrate that the lead ASO cocktails inhibit NMD of *CFTR*-W1282X mRNA by preventing the binding of EJCs located downstream of the PTC, beyond the footprint of the stalled ribosome. The lead GAIN ASO cocktails inhibited NMD of *CFTR*-W1282X mRNA by targeting presumptive EJC binding sites downstream of the PTC, independently of cell type. They efficiently suppressed NMD of *CFTR*-W1282X mRNA only when all downstream EJC binding sites that contribute to NMD (i.e., those on exons 24-26) were targeted by the ASOs. On the other hand, the lead ASO cocktails did not affect the total mRNA levels of *CFTR* alleles without nonsense mutation (F508del and G551D) or *CFTR* alleles with nonsense mutations upstream of exon 23 (G542X and R1162X). The lead ASO cocktails did not inhibit NMD of other endogenous NMD-sensitive transcripts we tested, or affect the levels of NMD-insensitive *CFTR* mRNA. These observations rule out the possibility of global NMD suppression due to ASO treatment, or *CFTR* mRNA stabilization by inhibition of other mRNA-degradation pathways. These results also show that the GAIN ASO cocktail’s effect inhibiting NMD, and thus increasing *CFTR* protein and chloride-transport activity, is specific for the W1282X allele, even though unintended exon-skipping occurred irrespective of the *CFTR* alleles. Based on the mechanism, a subset of the ASOs in the GAIN ASO cocktail might also be used to inhibit NMD of *CFTR* mRNAs with very rare nonsense mutations in exons 24 (e.g., Q1313X) or 25 (e.g., Q1330X and E1371X).”

Reviewer #3 (Remarks to the Author):

Low *CFTR* mRNA expression due to nonsense-mediated mRNA decay (NMD) is a major hurdle in developing a therapy for cystic fibrosis (CF) caused by the W1282X mutation in the *CFTR* gene. *CFTR*-W1282X truncated protein retains partial function, so increasing its levels by inhibiting NMD of its mRNA will likely be beneficial. Because NMD regulates the normal expression of many genes, gene-specific stabilization of *CFTR*-W1282X mRNA expression is more desirable than general NMD inhibition. Synthetic antisense oligonucleotides (ASOs) designed to prevent binding of exon junction complexes (EJC) downstream of premature termination codons (PTCs) attenuate NMD in a gene-specific manner. The authors developed a cocktail of three ASOs that specifically increases the expression of *CFTR* W1282X mRNA and *CFTR* protein in ASO-transfected human bronchial epithelial cells. This treatment increased the *CFTR*-mediated chloride current. Overall, this study sets the stage for clinical development of an allele-specific therapy for CF caused by the W1282X mutation.

The authors have addressed all of my prior concerns. Congratulations on this nice study!

Authors response> Thank you for the helpful feedback.

1. Valley H.C., *et al.* Isogenic cell models of cystic fibrosis-causing variants in natively expressing pulmonary epithelial cells. *J Cyst Fibros* **18**, 476-483 (2019).
2. Aksit M.A., *et al.* Decreased mRNA and protein stability of W1282X limits response to modulator therapy. *J Cyst Fibros* **18**, 606-613 (2019).
3. Haardt M., Benharouga M., Lechardeur D., Kartner N., Lukacs G.L. C-terminal Truncations Destabilize the Cystic Fibrosis Transmembrane Conductance Regulator without Impairing Its Biogenesis. *Journal of Biological Chemistry* **274**, 21873-21877 (1999).
4. Pinto M.C., Silva I.A., Figueira M.F., Amaral M.D., Lopes-Pacheco M. Pharmacological Modulation of Ion Channels for the Treatment of Cystic Fibrosis. *Journal of Experimental Pharmacology* **Volume 13**, 693-723 (2021).
5. Moschos S.A., Usher L., Lindsay M.A. Clinical potential of oligonucleotide-based therapeutics in the respiratory system. *Pharmacol Ther* **169**, 83-103 (2017).
6. Sermet-Gaudelus I., *et al.* Antisense oligonucleotide eluforsen improves CFTR function in F508del cystic fibrosis. *Journal of Cystic Fibrosis* **18**, 536-542 (2019).
7. Pharmaxis. Pharmaxis completes phase II clinical study with ASM8 in asthma patients. <https://www.pharmaxis.com.au/assets/Documents/pdf/2012/ASX/2012-04-17-ASM8-207-results.pdf>. Access Date: February 18, 2022. (2012).

REVIEWERS' COMMENTS

Reviewer #2 (Remarks to the Author):

The authors have addressed or explained all the issues raised.